# The undulating tripod gait as a model of the locomotion of walking fish

Michael Ishida ®[1] ✉, Fidji Berio ®[2], Theodora Po[2], Narges Khadem Hosseini[1], Neil H. Shubin[3], Valentina Di Santo ®[2] & Fumiya Iida ®[1,4]

A large subset of fishes capable of terrestrial walking exhibit strikingly similar gaits despite spanning across the phylogenetic space and having substantial differences in morphology. This recurrent pattern suggests the existence of shared mechanical principles underlying locomotor convergence. To investigate these principles, we analyze a common strategy we term the *"undulating tripod gait"*, a coordinated pattern of axial body undulation coupled with alternating anterior contact with the ground. In this work, we model the undulating tripod gait by approximating a fish's axial undulation as three rigid segments rotating with respect to each other and representing the anterior contact as a rigid beam that alternates the contact with the surface on the left and right sides of the body. Here, we focus on the grey bichir, *Polypterus senegalus*, as a specific exemplar of the undulating tripod gait. We perform high-speed kinematic analyses of terrestrial locomotion to identify baseline gait parameters, which we then validate as broadly representative by comparing them to those of other distantly related walking fishes. Using these parameters, we simulate the model to explore how variations in morphology and joint kinematics influence forward progression, revealing that peak locomotor performance emerges under conditions closely matching those observed in *P. senegalus*. Finally, we translate the model into a physical robot, demonstrating that the same simple coordination of axial undulation and anterior contact produces effective forward locomotion in the real world. By capturing core mechanical features shared across morphologically diverse species, this framework advances our understanding of terrestrial walking in fishes and offers a mechanistic lens through which to examine the evolutionary origins of locomotion in early vertebrates.

A wide range of extant fishes, including cartilaginous (chondrichthyans), lobe-finned (sarcopterygians), and ray-finned groups (actinopterygians), are capable of walking locomotion[1,2]. These fishes span a wide range of morphologies, including dorsoventrally compressed forms such as skates[3], elongate species like bichirs[4], and fishes with specialized fins such as mudskippers[5]. Understanding fish locomotion requires more than morphological analysis; performance arises from the dynamic interplay between body structure and environmental interaction[6]. On land, walking fishes generate forward motion through axial undulation (motion of the vertebral column),

[1]Department of Engineering, University of Cambridge, Cambridge, UK. [2]Scripps Institution of Oceanography, University of California San Diego, La Jolla, CA, USA. [3]Department of Organismal Biology and Anatomy, University of Chicago, Chicago, IL, USA. [4]Department of Precision Engineering, Graduate School of Engineering, The University of Tokyo, Tokyo, Japan. ✉e-mail: myi20@cam.ac.uk

appendicular articulation, or a coordinated combination of both, relying on ground reaction forces rather than hydrodynamic thrust[7]. Despite this shared reliance on substrate interaction, species differ markedly in their terrestrial walking strategies. Mudskippers use a "crutching" gait in which they push their bodies upward and forward using a synchronized motion of their pectoral fins[8], while epaulette sharks perform a "stepping" motion by articulating their pectoral and pelvic fins in sequence[9]. In contrast, catfishes combine lateral undulation and rolling of their bodies with a small amount of pectoral fin articulation[10].

Despite striking morphological differences, many fishes exhibit convergent patterns of undulatory locomotion during both walking and swimming. Rather than falling into discrete kinematic categories, swimming behaviors form a continuum shaped by the interplay of body form, flexibility, and flow characteristics[11,12]. However, with regards to terrestrial locomotion, previous work has instead attempted to classify walking fish using discrete categories such as using axial-bending, appendage-based support, or an axial-appendicular combination[7]. Of these, a few species employ almost entirely appendage articulation (e.g., mudskippers[13]) or almost entirely axial motion (e.g., ropefish[14]), but the vast majority of fishes exhibit both to a significant degree. It is difficult to further divide this main group into meaningful categories because of the extensive variation in locomotory kinematics between species. Species also exhibit different usages of their vertebral columns and pectoral fins depending on the environment in which they are moving[14,15], introducing further combinations of these characteristic motions. Classifying the gaits of different species in this manner, using these axial-appendicular categorizations, relies on kinematic observations without exploring the underlying physics principles of the gait.

Bioinspired robotics provides a powerful framework for probing the mechanics of walking locomotion in fishes[16] and for generating new biomechanical hypotheses about organismal function[17]. Robots inspired by animals can be used as stand-ins for the biological organisms to make counterfactual predictions about motions that are not directly observed in nature or to obey the ceteris paribus principle, in which testing the effects of a single feature can only be done while keeping the other features constant[18]. Models inspired by animals can range from anatomically detailed reconstructions that replicate species-specific morphology[19] to simple abstractions that isolate essential features[20]. By abstracting away taxon-specific traits, models can be built around shared mechanical features, enabling the identification of general principles[21] and the development of unifying frameworks across broad phylogenetic and temporal scales[22].

While fish swimming[23,24] and tetrapod walking[25] have been well-studied by roboticists, the functional morphology of walking fishes remains comparatively underexplored[26,27]. One type of walking locomotion is a combination of anchoring or propping the body with an anterior body part and then pushing the body around that anchor point using articulation of the axial skeleton. We call this the "undulating tripod gait" and we posit that this convergent gait (a gait evolved separately in a number of distantly related species) is exhibited by several species of amphibious fishes that belong to distinct taxonomic groups (Fig. 1). The undulating tripod gait is different from other fish gaits that generate locomotory forces primarily from their appendages rather than from axial articulation[8] as well as well-studied terrestrial walking gaits such as those of tetrapods[28,29], insects[30,31], and snakes which slither via complex frictional interactions with the ground[32].

In this work, we first focus on the locomotion of the gray bichir *Polypterus senegalus*, a ray-finned fish that can swim in water but also breathe and walk in air, as a specific exemplar of the undulating tripod gait. We introduce a mechanics-based model of the undulating tripod gait observed in different species of walking fishes. We measured the kinematics of the gaits of *P. senegalus* and several other species of walking fish to inform the model, which reduced the undulating body down to three rigid segments connected by actuated revolute joints and reduced the articulating fins to alternating contacts with the ground on the sides of the body. Then, we created a simulated robot based on the model to investigate how altering different parameters of the gait and the body geometry affected locomotion. Finally, we used a physical robot to investigate how realistic contact between substrates and the robot affected the walking motion.

## Results

### Definition of the undulating tripod gait model

The undulating tripod gait can be described as a coordinated combination of two motions: a propping action generated by an anterior body part, such as a pectoral fin, and a pushing action produced by the posterior body, including the axial musculature and caudal region. During locomotion, the fish alternates which side of the body makes contact with the substrate, using ground reaction forces to rotate its body around the planted prop. When the body is propped on the left side of the midline, the fish swings around the pivot point in the counterclockwise direction and vice versa when the body is propped on the right side of the midline (Fig. 2). The time series of postures within a gait cycle depends on the exact morphology and gait parameters; rather, it is the sequence of these general actions that defines the undulating tripod gait (Fig. 1).

To explore the principles of the undulating tripod gait, we created a simple theoretical model that recreates these salient features. In this model (Fig. 2), we represent the propping motion as being generated by a rigid beam rotating around the fish's head-tail axis ($\theta_1$) such that $\theta_1 > 0$ pushes the beam into the surface on the left side of the body. We approximate the continuum soft body as three rigid beams (lengths $L_1$, $L_2$, and $L_3$ from head to tail, respectively) connected by two revolute joints ($\theta_2$ between the head and center links and $\theta_3$ between the center and tail links) whose axes are both normal to the plane of the substrate on which the fish is walking. The rotations of $\theta_2$ and $\theta_3$ create the pushing motion that swings the body around the pivot point engaged with the surface. We refer to this as an *undulating tripod gait* because, under the model's kinematic constraints, the fish maintains contact with the ground at three key points: the tip of the propping beam and two locations along the body segments.

The precise positions of these contact points are not fixed but instead emerge from the fish's body shape and mass distribution. In this model, locomotion results only from the interplay of gravitational forces, joint trajectories, and friction-dominated contact with the substrate.

### Analysis of biological inspirations

In this work, we focus specifically on the terrestrial locomotion of *Polypterus senegalus* as an exemplar of the undulating tripod gait, in which the fish uses its pectoral fins as propping appendages (Supplementary Movie 1). *P. senegalus* is a species of fish capable of surviving in both aquatic and terrestrial environments, and its ability to live on land for extended periods of time has led to a significant literature on its terrestrial walking gaits[27,33,34]. Furthermore, its long, slender, flexible body is capable of significant axial bending, which potentially provides a larger feasible space of possible gait patterns compared to fish with stiffer axial columns.

We tracked the motion of six specimens across at least one complete gait cycle, defined as beginning and ending with the tail at its maximal rightward bend, to extract key parameters of the gait and body morphology for use in simulations and physical modeling. To approximate *P. senegalus* within the framework of the undulating tripod gait model introduced above, we digitized the midline of the body over time (Fig. 3a) and fit it using three connected linear segments (see "Methods" for more details). Based on observations of the walking motion, the lengths of these segments followed a 2:4:5 ratio, corresponding to 0.182, 0.367, and 0.455 of the body length (BL). With this

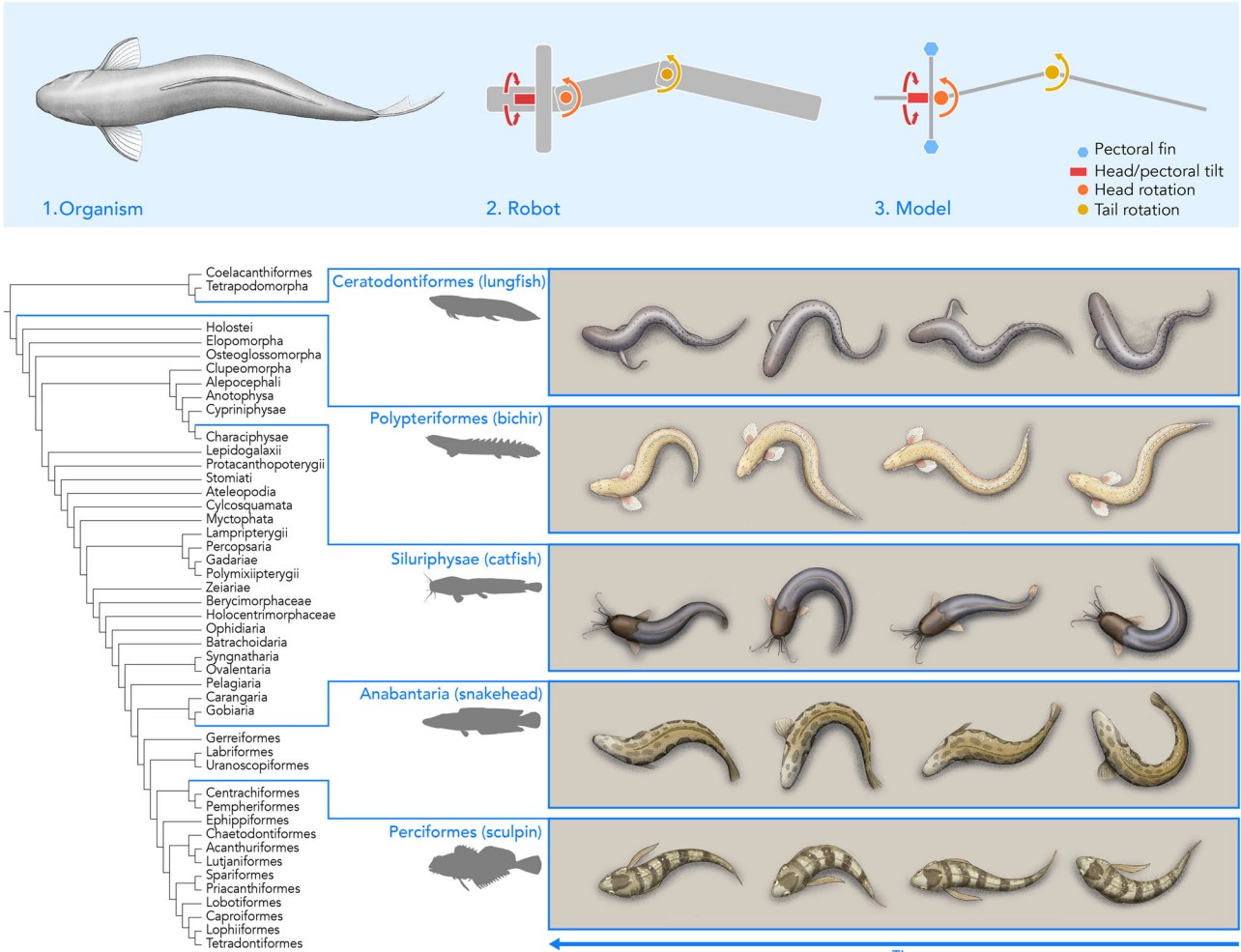

**Fig. 1 | Undulating tripod gait, as a result of convergent evolution, is reported in distant-related fish taxa.** Although body morphologies and kinematics slightly differ across species, all share a similar gait pattern in terrestrial environments. We model this gait as a coordinated motion of three body segments and alternating anterior contacts with the surface. The sketches shown are of representative body shapes taken throughout one gait cycle to illustrate the visual similarity of the kinematics of the following fish species: the African lungfish *Protopterus annectens*[35], the gray bichir *Polypterus senegalus*, the armored catfish *Hoplosternum sp.*[36], the northern snakehead *Channa argus*[38], and the tidepool sculpin *Oligocottus maculosus*[39].

ratio, the first joint was positioned near the pectoral girdle, as we observed little bending to the anterior of that point, and the second joint was placed slightly ahead of the pelvic girdle to capture both bending in the body and in the tail. Using this discretization, we observed that the angular motions at joints $\theta_2$ and $\theta_3$ exhibited approximately sinusoidal trajectories during periods of steady locomotion. Although gait frequency varied across individuals, each trial contained intervals of consistent timing and waveform shape sufficient for reliable sinusoidal fitting (Fig. 3e, f). The resulting amplitude and phase parameters are reported in Table 1 and served as input for both the simulated and physical robot models.

We found that this general gait pattern describes the walking locomotion of a diversity of fish species with different morphologies including those we highlighted in Fig. 1. We tracked the midlines (see "Methods" for more details) of five different fish that exhibit the undulating tripod gait: *P. senegalus*, the African lungfish *Protopterus annectens*[35], several species of catfish *Pterygoplichthys disjunctivus, Pterygoplichthys multiradiatus, Pterygoplichthys gibbiceps, Hoplosternum punctatus, and Hoplosternum sp.*[36,37], the northern snakehead *Channa argus*[38], and the tidepool sculpin *Oligocottus maculosus*[39] (Fig. 4a). We split the continuous walking behavior into individual gait cycles starting and ending with the tail to the right side of the fish's body and averaged the per-cycle observations. We took the anterior

point of the head as the primary propping point for *P. annectens* and the pectoral fins as the primary propping points for the other species. Across species, all the fish alternate which side of the body makes contact with the substrate and subsequently sweep the tail toward that planted side (i.e., after engaging the left pectoral fin, the tail bends leftward). Although the timing of individual movements and the location of the propping structure vary among species and specimens, the overall sequence of prop, bend, and pivot remains consistent within the undulating tripod gait. The gait pattern is generally symmetric (Fig. 4e).

Similarly, when these species were approximated using the same discretization into three rigid segments (see "Methods" for more details), the trajectories of $\theta_2$ (Fig. 4b) and $\theta_3$ (Fig. 4c) exhibited consistent sinusoidal patterns. While the amplitude of joint motion varied among species, the phase relationship between the time series trajectories of $\theta_2$ and $\theta_3$ remained conserved. This is strong evidence that the walking gaits of these species can all be described by the undulating tripod gait model, with *P. senegalus* serving as an exemplar.

## Simulating morphology and motion

To systematically evaluate the locomotion of a system under the constraints imposed by our model, we created a simulation of a robot that uses the undulating tripod gait. The simulation is inspired by the

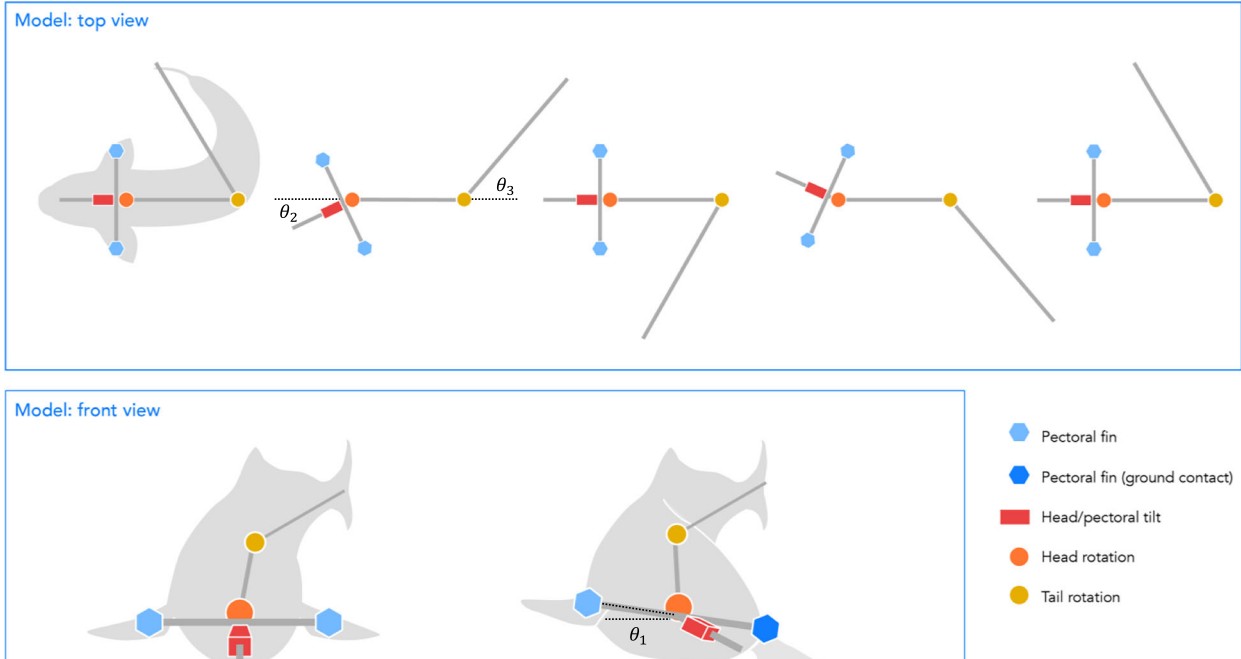

**Fig. 2 | The simplified model of the undulating tripod gait, where the soft body of the fish is discretized into three rigid segments that rotate with respect to each other.** The blue hexagons indicate the pectoral fins, where the dark blue specifically indicates contact with the ground. The red box indicates the head and pectoral tilt $\theta_1$ (rotation in roll, aligned in the anterior-posterior direction), the orange circle indicates the head rotation $\theta_2$ (rotation in yaw, aligned in the dorsal-ventral direction), and the yellow circle indicates the tail rotation $\theta_3$ (rotation in yaw, aligned in the dorsal-ventral direction). The schematics are not linked to specific timesteps, as the postures of the fish within its gait cycle depend on the exact gait parameters.

morphology and gait kinematics of *P. senegalus* and subject to the approximations made in our model of the undulating tripod gait. We used this simulation to investigate parameters of both the morphology and the gait that were not observed in our experiments with real fish and to determine how these parameters affect walking locomotion (see "Methods" for more details).

In our simulation, the fish's body was represented by three rigid links that rotate with respect to each other in the same plane via revolute joints. From the tracking of the fish described in Fig. 3, the head, middle, and tail segments have lengths of a 2:4:5 ratio, with an overall length of 0.1 m, similar to the length scale of *P. senegalus*. These joints were controlled by time-series angular position inputs, and the robot is driven forward by the interactions between the robot and the ground, calculated by the simulation environment. Here, we assume that there is sufficient capacity for torque generation in each of the joints to satisfy the angular positions over time. The simulated model only allows for rotation of the body joints in the frontal plane, although the fish can bend its body in 3D space. Since this simulation is inspired by *P. senegalus*, we represent the propping appendages as a rigid bar at the base of the head segment, approximating the pectoral fin contacts; to model other species using different props, this structure would need to be adjusted.

We used a generic contact model consisting of a spring and damper (normal force preventing the body from penetrating the substrate) and static and kinetic friction (shear force in the plane of the substrate), and these forces, combined with the posture and mass distribution of the robot, determined the location of the contacts between the body and the substrate.

Because the observed bending of the fish's body can be approximated by time series sinusoidal joint positions (Fig. 3e, f), we imposed sinusoidal inputs to the actuators of the simulated robot. We approximated the behavior of the fin joint as a saturating sinusoid, where the joint reaches a threshold position and holds a constant angular position analogous to the phase of the gait where the fish is not moving its pectoral fin that is in contact with the ground. We used the parameters observed in the fish's walking motion described in Table 1 to set a baseline for the simulated robot's walking motion.

## Altering the gait of the simulated robot

To examine the effects of individual gait parameters on locomotion, we varied the parameters of the sinusoidal inputs (amplitude, frequency, and phase difference between the sinusoids) to the joints from the baseline simulation parameters that were based on observations of *P. senegalus*. Increasing the frequency of the joint oscillation is analogous to the fish undulating its body at a higher frequency; increasing the amplitude is analogous to the fish bending its body to a greater maximum curvature.

**Increasing the frequency of undulation increases locomotion speed.** The frequency of the joints' oscillations (i.e., how quickly the undulation happens) affects the relative velocity of each segment and the number of steps the robot takes per second. Here, we simulated the locomotion of the robot at frequencies of 0.5, 1.5, 2.5, 3.5, and 4.5 Hz while holding the other parameters of the sinusoids constant as defined in Table 1. This range encompasses the frequencies that were recorded in our experiments with the real *P. senegalus* specimens.

The walking speed of the simulated robot increased linearly as the frequency of the sinusoidal input to the motors increases, but the distance traveled per cycle decreased as the frequency increases. The measured speed of *P. senegalus* was slower than that of the simulated robot, although the discrepancy between the speed of the simulated robot and the real fish was smaller at higher frequencies than at lower frequencies (Fig. 5a). The lower distance per step indicates an increased amount of detrimental slipping between the robot and the

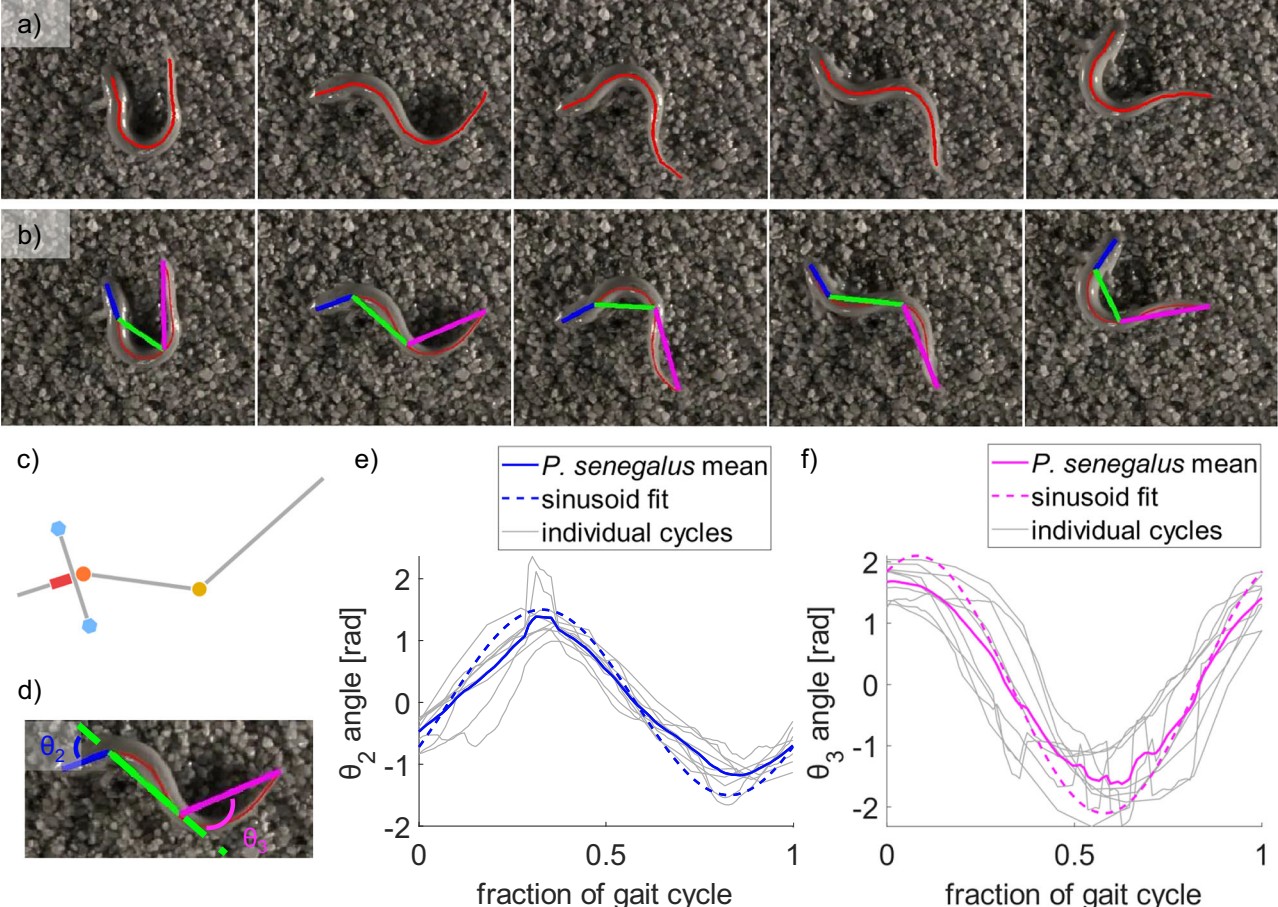

**Fig. 3 | Tracking of the walking motion of *Polypterus senegalus*. a** Time series images of one cycle of *P. senegalus*'s gait with the tracked midline overlaid (red). **b** Time series images of one gait cycle with the discretization of the body into three linear segments overlaid. The three-segment discretization is denoted by blue (head segment), green (body segment), and pink (tail segment) lines. Scale bar is 5 cm. **c** Comparison of the theoretical tripod gait model to **d** the three-segment discretization. **e** Fitting a sinusoid to $\theta_2$, the joint between the head and body segments, and **f** fitting a sinusoid to $\theta_3$, the joint between the body and tail segments, where each trial is plotted in gray, the average of the four trials is plotted in solid color, and the fitted sinusoid is plotted as a dashed line.

### Table 1 | Fitted sinusoidal parameters identified from the observations of *P. senegalus*

| Parameter | Symbol | Fitted value [rad] |
| --- | --- | --- |
| amplitude of the head sinusoid | $A_2$ | 1.5 |
| phase of the head sinusoid | $\phi_2$ | $\pi/2$ |
| amplitude of the tail sinusoid | $A_3$ | 2.1 |
| phase of the tail sinusoid | $\phi_3$ | $\pi$ |

ground as the frequency increased. Increasing the frequency of the sinusoids increases the number of steps that the robot takes per unit time, and the additional steps compensate for the shorter individual steps (Fig. 5b).

**Varying amplitude of the body joints' movement changes locomotion speed.** The amplitudes of the joints' oscillations (i.e., how much the midline of the body bends) can also significantly affect the locomotion speed of the simulated robot. Here, we simulated the locomotion of the robot at frequencies of 1.5, 2.5, and 3.5 Hz, varying the amplitude of the sinusoid applied to one of the body joints while keeping the other parameters of the sinusoids constant as defined in Table 1. We took advantage of the virtual environment to simulate joint motions that would create self-collision in the robot; they are feasible states for the fish because its soft body bends along the continuum,

but not for a rigid robot with discrete joints between rigid elements. To eliminate the effects of self-collisions in the simulation, we only defined contact forces between the robot and the substrate and not between the body elements.

The simulated robot moved fastest with the amplitudes of $\theta_2 = 1.5$ and $\theta_3 = 2.1$, which are the same gait parameters observed in our experiments with *P. senegalus* specimens. The robot's forward speed increased with the amplitude of the oscillation of the head joint until a maximum of $\theta_2 = 1.5$ rad and then reduced with further increase of amplitude (Fig. 6a). The robot's speed also increased as the amplitude of the tail joint increased up to the maximum physically achievable angle $\theta_3 = 2.1$ rad and then reduced with further increase of amplitude achievable in simulation only (Fig. 6b).

Varying the amplitude of $\theta_2$, the angle between the head segment and the middle segment, significantly affects the forward speed of the robot (Supplementary Movie 2). Because the fins are attached to the head joint, the amplitude of $\theta_2$ affects where the fin is planted with respect to the rest of the body. From the kinematics of the tripod gait, we expect that the most effective gait will be when the limits of the head joint $\theta_2$ oscillation are at $\pi/2$ and $-\pi/2$. If the fin does not slip against the ground, the head will move in an arc around the stationary fin. Then, the maximum possible step size is where the fin is planted directly in front of the robot in the direction of motion. The joint then rotates $\pi$ radians and plants the other fin directly in front of the robot.

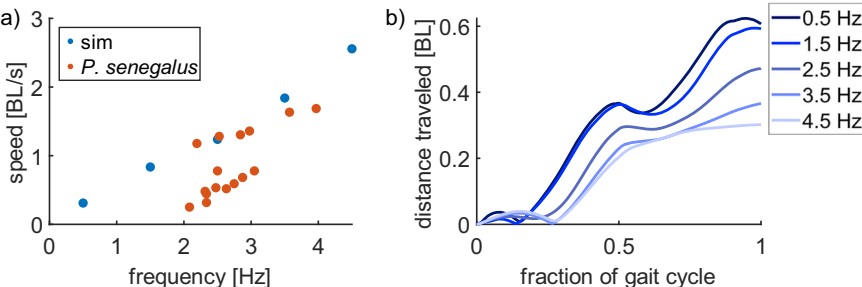

**Fig. 4 | Comparison of the walking kinematics of five fish species exhibiting the undulating tripod gait. a** Illustrations of the five species - the African lungfish *Protopterus annectens*, the gray bichir *P. senegalus*, several species of catfish *Pterygoplichthys disjunctivus, Pterygoplichthys multiradiatus, Pterygoplichthys gibbiceps, Hoplosternum punctatus,* and *Hoplosternum sp.*, the northern snakehead *Channa argus*, and the tidepool sculpin *Oligocottus maculosus*. **b** Angular position of the first body joint over the course of one gait cycle for each of the fish species. **c** Angular position of the second body joint over the course of one gait cycle for each of the

fish species. The lighter traces denote each gait cycle tracked, and the dark, bolded traces are the average of the tracked gait cycles. The number of specimens for each are $n = 2$ (lungfish), $n = 6$ (bichir), $n = 5$ (catfish), $n = 4$ (snakehead), $n = 1$ (sculpin). **d** Diagram of the discretization overlaid on an illustration of the bichir, where the dotted line denotes an extension of the middle segment to define the angles $\theta_2$ and $\theta_3$. **e** Gait diagram of the walking locomotion of the five species, divided into the movement of the tail and the propping anterior contact between the fish and the surface. More details can be found in the Supplementary Information.

**Fig. 5 | Effects of the frequency of the sinusoids applied to the joints on simulated robot locomotion. a** Speed of the simulated robot varying frequency (all joints operate at equal frequencies within the trial). **b** Distance traveled during

one gait cycle for each frequency. In all cases, the other gait parameters were held constant and matched those tracked from *P. senegalus*: $A_2 = 1.5$, $A_3 = 2.1$, $\phi_2 = \pi/2$, $\phi_3 = \pi$.

With three rigid body segments, the tail segment contacts the ground and creates most of the pushing force, so the oscillation of the tail joint $\theta_3$ affects both the magnitude and direction of the ground reaction force. As the amplitude of the tail joint increases,

both the range of motion and the relative velocity of the tail segment increase because of the larger angular sweep at the same frequency (Supplementary Movie 3). However, at the highest values of tail amplitude, the angle of the tail drives forces that are more

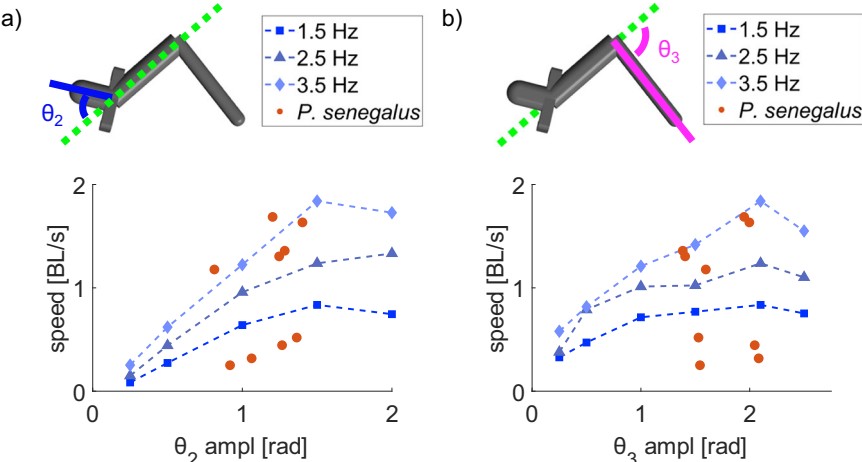

**Fig. 6 | Effects of the amplitude of the body joints on simulated robot speed.**
**a** Speed of the simulated robot varying the amplitude of the head joint ($\theta_2$) oscillation at different actuation frequencies. **b** Speed of the simulated robot varying the amplitude of the tail joint ($\theta_3$) oscillation at different actuation frequencies. In all cases, we held the other gait parameters constant, matching those tracked from *P. senegalus*: $\phi_2 = \pi/2$, $\phi_3 = \pi$. When varying the amplitude of $\theta_2$, we held $A_3 = 2.1$; when varying the amplitude of $\theta_3$, we held $A_2 = 1.5$.

horizontal than forward, reducing the effectiveness of the additional tail sweep.

**Phase differences between joint actuation affects direction of locomotion.** Increasing the frequency or amplitude of the sinusoids affects the speed of the robot but does not affect the sequence of body shapes that are generated. To investigate the gait space of the simulated robot, we varied the phases of the actuated joints to explore how the sequence of joint actuation affected locomotion with the same frequency (2.5 Hz) and amplitude (1.5 rad and 2.1 rad for the head and tail joints, respectively). The joint controlling the fin actuation $\theta_1$ has the same phase in all the simulations, where both fins begin off the surface, but the joint then rotates such that the left fin contacts the surface first. When the phase of the head joint is zero ($\phi_2 = 0$), the head segment begins in-line with the middle segment and begins by rotating clockwise with respect to the joint (i.e., the head rotates to the right of the body). When the phase of the tail joint is zero ($\phi_3 = 0$), the tail segment begins in-line with the middle segment and begins by rotating clockwise with respect to the joint (i.e., the tail rotates to the left of the body).

During observations of *P. senegalus* specimens, we noticed that the coordination between the pectoral fin contact and the tail motion differed between forward and backward walking (Fig. 7a and Supplementary Movie 4). This was also reflected in the simulations varying phase, where the robot consistently moved forward when $\phi_2 = 0$ or $\phi_2 = \pi/2$ and consistently walked backward when $\phi_2 = \pi$ or $\phi_2 = 3\pi/2$. In addition, the simulated robot moved forward fastest when $\phi_2 = \pi/2$ and $\phi_3 = \pi$, which were the same parameters observed in the fish's locomotion (Fig. 7b, c).

When $\phi_2 = \pi/2$, the robot plants its left fin on the ground and rotates its head counterclockwise before planting its right fin. With this motion, the right fin then ends up forward of its starting position. The reverse is true when $\phi_2 = 3\pi/2$; the robot plants its left fin and rotates its head clockwise, and the right fin ends up behind its starting position. This indicates that the phase of the head joint with respect to the fin actuation can determine whether the robot walks forward or backward. When $\phi_2 = 0$ or $\phi_2 = \pi$ (i.e., the head is in line with the center segment as it changes which fin is planted), the simulated robot's speed is greatly reduced, in some cases barely producing net locomotion in any direction (Fig. 7b).

The effects of $\phi_3$ on the overall speed and direction of the simulated robot are more difficult to categorize. However, the maximum

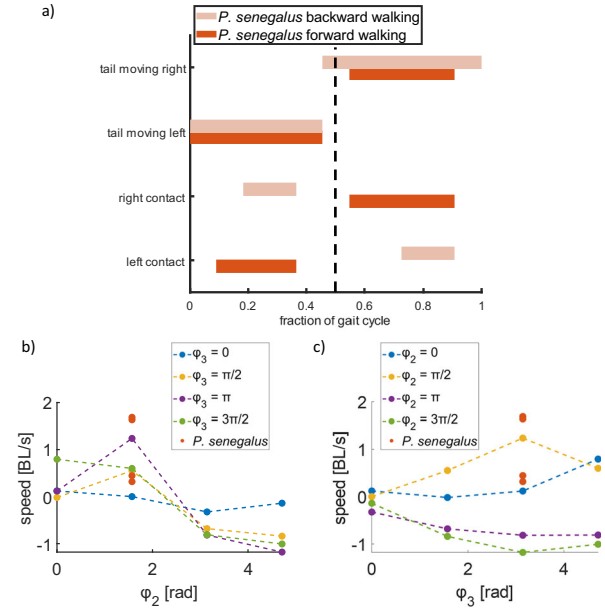

**Fig. 7 | Effect of phase difference between the sinusoids applied to the joints on the direction of locomotion. a** Gait diagram of *P. senegalus* for observations of forward and backward walking. **b** Speed and direction of the simulated robot for different phase offsets as a function of $\phi_2$. **c** Speed and direction of the simulated robot for different phase offsets as a function of $\phi_3$.

forward and backward walking speeds were achieved with $\phi_3 = \pi$, while $\phi_3 = 0$ led to the worst locomotion (Fig. 7c). This indicates the importance of the coordination of the tail and pectoral fin during locomotion. When $\phi_3 = \pi$, as the left pectoral fin comes in contact with the surfaces, the tail is bending to the right side of the body and then sweeps to the left; when $\phi_3 = 0$, the opposite is true.

**Varying morphological parameters.** The previous experiments were all done with a simulated robot at a similar scale to the biological inspiration, *P. senegalus*. However, we have observed the undulating tripod gait in a number of other species with different body shapes and sizes (Fig. 4), so we investigated the performance of our model with a variety of morphologies. Although we expect the exact details of

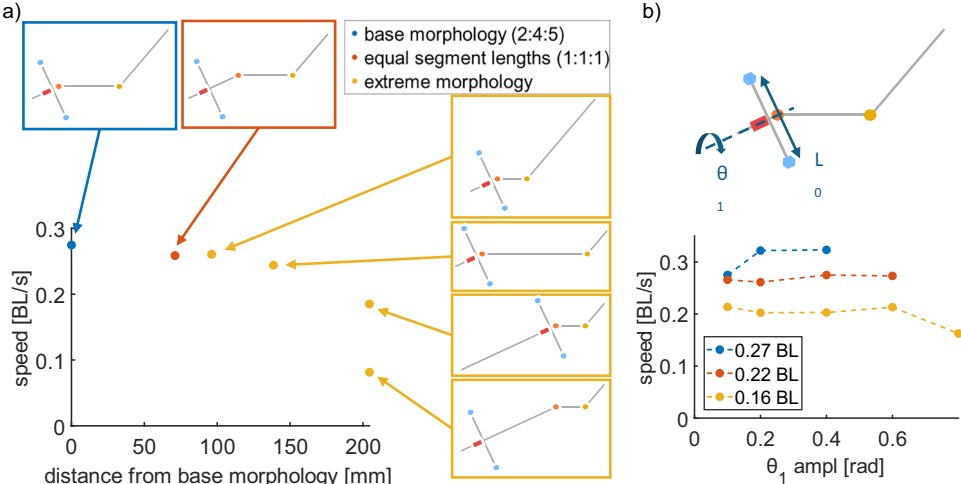

**Fig. 8 | Exploring the effects of body morphology on the locomotion of the simulated robot. a** Speed of the simulated robot with the same total body length but different body proportions. Blue denotes the base morphology (2:4:5 ratio of segment lengths), red denotes the morphology with equal segment lengths (1:1:1 ratio of segment lengths), and yellow denotes the morphologies with two short segments and one long segment (either 2:2:7, 2:7:2, or 7:2:2 ratios of segment lengths). Details of the calculation of the Euclidean difference between the altered morphologies and the base morphology can be found in the "Methods" Section. **b** Speed of simulated robot versus amplitude of the actuated fin segment for different lengths of the fin segment $L_0$. Each version of the simulated robot was tested with five amplitudes of $\theta_1$; points not included in the plot were cases in which the robot could not produce stable locomotion and rolled.

locomotion applying the undulating tripod gait to different morphologies to be slightly different, our formulation of the undulating tripod gait model should be independent of length scale and mass scale.

To explore the effects of scale on the undulating tripod gait, we increased the size of the simulated robot from 0.1 to 0.37 m and increased its mass from 10 g to 500 g, and applied the same gait observed in the fish experiments (Table 1). For this set of simulations, we reduced the frequency of the gait to 0.5 Hz because of the larger body size with more inertia. These parameters were chosen to be of similar scale to a physical robot, and with these parameters, the simulated robot achieved a speed of 0.27 BL/s (Fig. 8a).

The formulation of our model of the undulating tripod gait includes the assumption that the body can be split into no less than three rigid beams, which have proportional lengths drawn from observations of *P. senegalus*. To assess the validity of these assumptions, we simulated the motion of the robot under different methods of discretizing the continuum body. We reduced the number of body segments from three to two by holding either (1) the head joint stationary ($\theta_2(t) = 0$), effectively combining the head and middle segments into a single rigid body, or (2) the tail joint stationary ($\theta_3(t) = 0$), effectively combining the middle and tail segments into a single rigid body (Supplementary Movie 7). The maximum forward speed of the simulated robot in either of these configurations with two body segments was 6.1% of the maximum forward speed with three body segments (Supplementary Table 1).

We also explored how the locomotion speed changed with different lengths of the three rigid beams (Fig. 8a). We considered the discretization based on observations of *P. senegalus* (2:4:5 ratio of the segment lengths) as the base morphology and explored two additional classes of morphologies with increasing variation from the base morphology while keeping the overall length of the simulated robot constant. The first variation was with all three segments of equal length (i.e., 1:1:1 ratio of the segment lengths such that each segment is 1/3 the total length of the base morphology). The second set kept two segments at the minimum lengths and varied which of the three segments the long segment (i.e., 2:2:7, 2:7:2, and 7:2:2 ratios of the segment lengths). The walking speed of the simulated robot with these morphologies is plotted against the difference (Euclidean distance)

between the new segment lengths and the base morphology. Schematics of the different morphologies are included in Fig. 8a.

All of the simulated robots with altered morphologies were capable of forward locomotion with the same sinusoidal control parameters applied in previous experiments (Supplementary Movie 5). This indicates that the model is valid for a wide range of morphologies, even including some that are implausible for real fish to exhibit. These example morphologies were chosen to span some of the potential design space of the simulated robot and support the idea that a range of species capable of various degrees of body bending would be expected to exhibit different parameters for the undulating tripod gait (Fig. 4).

Finally, we investigated the effects of the length of the fin segment $L_0$ that performs the propping function (Fig. 8b). As the length of the fin segment increased, the simulated robot had a higher speed because the length of each step was longer. However, the robot was also less stable in roll because the same amount of rotation of $\theta_1$ (angle of the fin segment with respect to the axis of the robot) with a longer fin segment caused the robot to rotate more in relation to the ground. For $L_0 = 0.27$ BL, the simulated robot rolled when the amplitude of the joint oscillation was 0.6 rad or higher; for $L_0 = 0.22$ BL (the base length), the simulated robot rolled when the amplitude of the joint oscillation was 0.8 rad or higher. Furthermore, the increased amplitude of $\theta_1$ tended not to cause the simulated robot to move faster, implying that additional rolling of the body does not generally affect the locomotion speed of the tripod gait.

## Physical robot

To verify that the conclusions drawn from the simulated robot can be extrapolated to the physical world, we built a robot with the same three-segment structure (Fig. 9a). We varied the amplitude of each of the two body joints at a constant frequency of 0.5 Hz.

Unlike in simulation, the potential for self-collisions between the rigid segments created kinematic constraints limiting the maximum amplitude of the head joint to $\theta_2 = 1.5$ and the maximum amplitude of the tail joint to $\theta_3 = 2.1$ (see "Methods" for more details).

Here, we found that the robot successfully moved forward (Supplementary Movie 6) and, like with the simulated robot, had the highest speed when moving with the gait parameters observed in *P.*

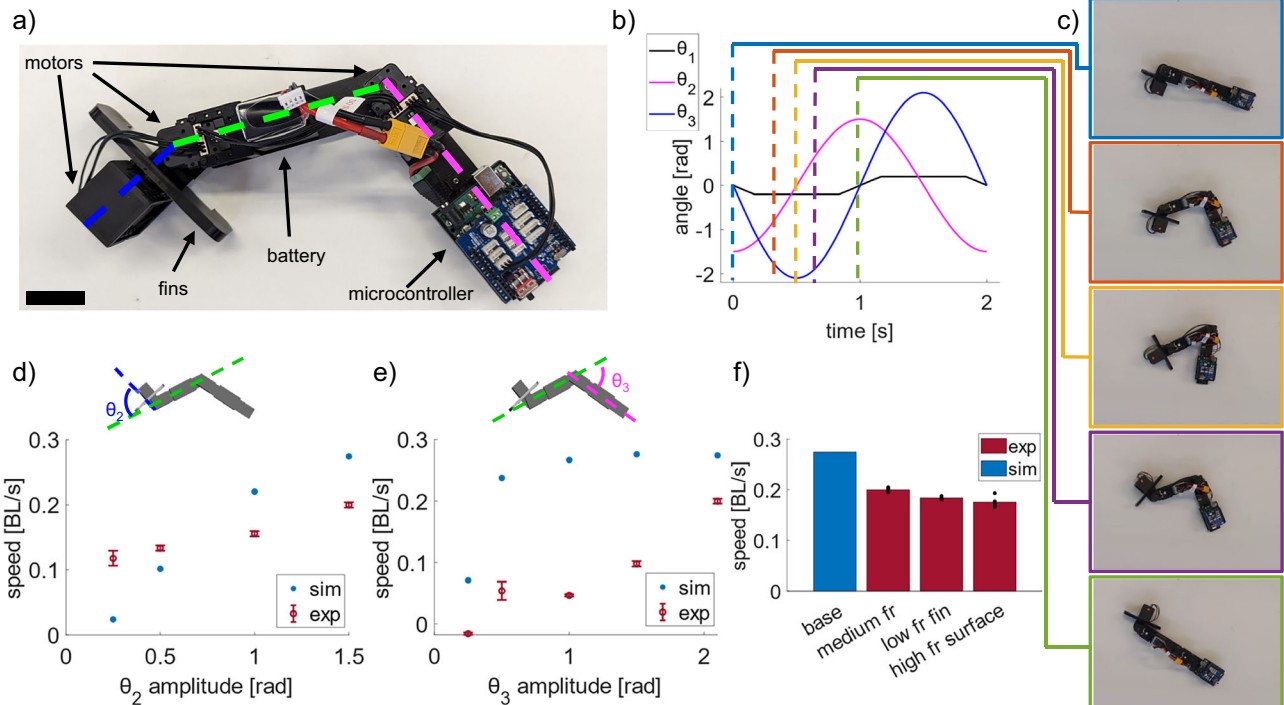

**Fig. 9 | Locomotion of the physical robot. a** Top view image of robot with scale bar 35 mm. **b** Plot of angle vs. time inputs to the three motors during one cycle of the base gait. **c** Time series images of the robot using the base gait for half of one gait cycle at timesteps corresponding with the annotated plot of control signals. **d** Comparison of the speed of the physical robot and the simulated robot, varying the amplitude of the head joint ($\theta_2$) oscillation. We held the other gait parameters constant, matching those tracked from *P. senegalus*: $\phi_2 = \pi/2$, $\phi_3 = \pi$, and $A_3 = 2.1$. **e** Comparison of the speed of the physical robot and the simulated robot, varying the amplitude of the tail joint ($\theta_3$) oscillation. We held the other gait parameters

constant, matching those tracked from *P. senegalus*: $\phi_2 = \pi/2$, $\phi_3 = \pi$, and $A_2 = 1.5$. **f** Comparison of the speed of the robot on different surfaces. "Base" denotes the robot with high friction fins on the low friction surface ($\mu_s = 0.30$), but the robot was not able to walk with the high friction fins on the high friction surface ($\mu_s = 0.38$). "Low fr fin" denotes the robot with low friction fins on the low friction surface ($\mu_s = 0.27$), and "high fr surface" denotes the robot with low friction fins on the high friction surface ($\mu_s = 0.32$). Error bars denote the standard deviation above and below the mean point for $n = 5$ trials.

*senegalus*. When varying the amplitude of the input to the head joint, the speed increased up to the maximum amplitude defined by its geometry (Fig. 9d). However, changing the amplitude did not change the speed of the physical robot as much as it changed the speed of the simulated robot. When varying the amplitude of the tail joint, the speed of the physical robot also increased as amplitude increased (Fig. 9e). The trajectory of the robot was consistent when keeping gait parameters constant (Supplementary Fig. 6), though the speed of the physical robot was much lower than the simulated robot until the highest amplitudes tested.

Because the contact model in the simulation was idealized compared to real surface contact, we performed further experiments to investigate how friction between the robot and substrate affected locomotion. We ran a set of experiments on a high friction (carpet) and low friction (synthetic table) substrates with a high friction fin tip (sandpaper) and a low friction fin tip (plastic) using the robot's base gait ($A_{\theta 2} = 1.5$ and $A_{\theta 3} = 2.1$) to explore the effects of surface roughness and heterogeneous body friction on locomotion (Fig. 9f). While we tested the robot with both high and low friction fins on both high and low friction surfaces, the robot was not able to move with high friction fins on the high friction surface as it consistently rolled over, indicating that some motion of the fin against the ground might be necessary for locomotion with our rigid robotic approximation. However, on the lower friction surface, the robot's walking speed was 8.5% faster with high friction fins instead of low friction fins. This dichotomy indicates the importance of managing the friction between the robot and the surface, as some degree of slippage between the robot and the surface is necessary to move forward, but too much slippage reduces the forward motion.

## Discussion

In this study, we introduced a minimal mechanical model of the undulating tripod gait, a locomotor strategy shared across phylogenetically diverse walking fishes. By abstracting the body into three rigid segments and representing fin-ground interactions as alternating point contacts, we demonstrated that forward terrestrial locomotion can emerge from simple axial bending and substrate reaction forces. Using both simulations and a physical robot, we explored how gait parameters such as bending amplitude, phase coordination, and morphology affect locomotor performance. Our results show that the gait observed in *Polypterus senegalus* occupies a region of the parameter space associated with peak locomotor speed, and that this basic coordination pattern is robust across a range of morphologies and scales. We also found that the direction of locomotion depends on the timing between head movement and fin placement, while the walking speed is modulated by both the amplitude of axial bending and the frequency of undulation. Notably, the gait remained effective across a wide range of morphologies, including configurations that are biologically implausible. Finally, we found that the fastest walking gait of both the simulated and physical robots derived from the morphology of *P. senegalus* aligned closely with the empirical kinematics observed in live animals.

The widespread presence of the undulating tripod gait among distantly related taxa of walking fishes with varying but overlapping morphologies highlights the extent of locomotor convergence in terrestrial fish locomotion[40,41]. This convergence is not likely due to shared ancestry, as evidenced by the different terrestrial walking strategies of closely related species such as *P. seneglalus* and *E. calabaricus* (both of the family Polypteridae), but rather due to common

mechanical constraints imposed by body–substrate interactions. The ability to exploit these interactions without substantial morphological innovation may reflect a broader functional plasticity in fishes via which latent terrestrial capabilities may be expressed under specific environmental conditions (e.g., hypoxic stress, predator evasion, seasonal habitat shifts, or movement between tidal pools) where temporary access to land or structured substrates can confer survival advantages[40,42]. Similar hypotheses have been posited regarding fishes that exhibit jumping behaviors without specialized anatomy[43], providing additional evidence that fish can generate different kinematics as needed on land.

Moreover, the simplicity of the gait may offer energetic advantages. By relying on passive ground reaction forces, stable axial resonance, and low-complexity joint trajectories, the undulating tripod gait minimizes the need for precise control of stepping or high force output from anterior structures (e.g., pectoral fins) compared with more specialized gaits[8,9]. This may be particularly advantageous in low-oxygen environments or during anaerobic bursts of activity, where reducing postural costs and maintaining mechanical efficiency is essential[44]. Benthic species not specialized for terrestrial locomotion could also benefit from the use of this gait in submerged environments as a more stable and energetically efficient means of moving slowly or station holding, as hovering in midwater incurs high energetic costs due to the need for constant fine motor control to counteract mechanical instability[45].

The undulating tripod gait offers a biomechanical bridge between aquatic undulation and the emergence of limb-driven terrestrial gaits in the water-to-land transition, potentially shaping how we reconstruct the locomotor ecology of early tetrapods[22,46]. Transitional fish species such as *Tiktaalik roseae* possessed robust pectoral girdles and flexible fins but lacked the fully developed appendicular structures needed for true stepping gaits[47–49]. It is plausible, however, that *T. roseae* and other related taxa that predate the evolution of digit-bearing limbs[50,51] would instead have been capable of the relatively simple axial-driven undulating tripod gait. Our findings also complement experimental and comparative studies that highlight the persistence of axial-based walking in living sarcopterygians such as lungfish[50] and developmental plasticity in *P. senegalus* reared in terrestrial environments[33].

The framework for modeling fish-like walking we presented here builds on existing bioinspired robotics studies that explore animal locomotion through physical embodiment and parameter manipulation[16,18,52,53]. By abstracting away species-specific anatomical complexity, we isolated the fundamental features of morphology necessary for producing forward terrestrial walking, and by approximating the gait as a set of sinusoids, we reduced kinematic noise from biological recordings to enable consistent exploration of the gait parameter space. The general undulating tripod gait can be made optimal to specific cases by selecting parameters based on details of each fish species (e.g., specific anatomical features) and its interaction with the environment (e.g., the friction between the skin and the substrate[54] or the terrain it walks on[14,15]).

Importantly, we demonstrated that the gait remains functional across a wide range of body shapes, stiffnesses, and actuation frequencies that are difficult or impossible to test in vivo, allowing us to formulate hypotheses on the effectiveness of gaits that might have arisen in other extant and extinct fish species. For example, fish with stiffer bodies could still produce the undulating tripod gait, but would likely walk more slowly if they do not achieve the same degree of axial bending. More broadly, this work is an example of a robotic model that can be used to explore counterfactual kinematic and morphological regimes or to reveal mechanically viable locomotor strategies that may be inaccessible due to developmental, physiological, or evolutionary constraints.

While the undulating tripod gait model captures the core biomechanical elements necessary for terrestrial locomotion in walking fishes, it also has notable limitations. Chief among these is the abstraction of the fish body into three rigid segments operating strictly in the sagittal plane, which excludes the nuanced 3D bending, twisting, and postural adjustments observed in real animals. Though the undulating tripod gait model is generalizable to many species, both the propping points and the segment lengths need to be adjusted to represent the mechanics of each individual species. While our simulations used kinematics derived from *P. senegalus*, we did not model internal dynamics such as muscle actuation, body elasticity, or metabolic constraints[55] that could substantially impact gait efficiency and feasibility in biological systems[12]. Furthermore, our model treats contact as a binary event governed by spring-damper mechanics and Coulomb friction, neglecting the compliant and dynamic contact often observed between pectoral fins and the surface[56]. This omission may underestimate the ability of the fish to actively modulate load distribution or to fine-tune ground reaction forces in response to environmental feedback[14,34]. Notably, our robot's locomotion required a balance between high enough friction for reacting forces and low enough resistance to permit anterior gliding. We achieved these conditions both under simplified substrate mechanics in simulation and in the physical world, though more work should be done to bridge these two environments.

Looking forward, future work should systematically investigate the sensitivity of the model to specific gait and morphological parameters of interest. In particular, further analysis of posterior body–substrate contact is needed, including quantifying the magnitude and direction of locomotory forces generated by the tail and determining whether pelvic fin contact constrains motion or alters ground reaction forces. It would also be valuable to integrate models of energy cost, substrate interaction, and environmental feedback to assess gait efficiency and robustness under ecologically relevant conditions, while explicitly accounting for real-world surface properties such as compliance, viscoelasticity, and fluid layering. Finally, extending the model to simulate transitions between aquatic and terrestrial environments could offer insights into how environmental pressures may have shaped early locomotor strategies during the vertebrate water-to-land transition.

In summary, this study presents the undulating tripod gait as a simple yet powerful framework for understanding terrestrial locomotion in phylogenetically distant species of walking fish. We show that forward locomotion can emerge across diverse body plans through axial flexibility, frictional interactions, gravity, and a conserved pattern of body–substrate coordination, without requiring the complex limb morphologies characteristic of tetrapods. By coupling biological observations with physical and computational robotic models, this approach highlights how minimal mechanical principles can be used to interpret both extant and extinct forms of locomotion[22]. This work represents a step toward identifying the ecological contexts in which walking provides a functional advantage over swimming[12,57], and how such pressures may have contributed to the early evolution of terrestrial locomotion in vertebrates.

## Methods

### Ethics statement

The experiments on the fish used in this study were performed under approved Marine Biological Laboratory IACUC animal care protocols (no. 11924-2020 and no. 23-19).

### Tracking the kinematics of the fish

We conducted experiments with four 1-year-old *Polypterus senegalus* specimens, the species selected as an exemplar of the undulating tripod gait, to quantify the kinematics of their terrestrial locomotion. Dorsal-view videos were recorded using high-speed cameras (Fastec H7 or Chronos 2.1, Krontech) at 250 fps as the fish walked across damp sand. A single gait cycle was defined as the interval between successive

instances of the tail reaching its maximal lateral displacement. Each video was downsampled to yield 11–13 evenly spaced frames per gait cycle. Within each frame, we extracted the midline of the body and fit a series of 200 equidistant points along its length, normalized to the total body length of the fish[11]. For videos containing *P. senegalus*, the positions of the snout, tail, and proximal and distal fin landmarks were tracked using DeepLabCut[58]. DeepLabCut is a markerless pose-estimation framework that employs a convolutional neural network to track user-defined anatomical landmarks. We trained a ResNet-50–based network[59] using twenty manually labeled frames per video, selected automatically using a combination of k-means clustering and uniform sampling. Data augmentation was performed using the "imgaug" framework[60]. The network was trained for 1,000,000 iterations using all videos included in the dataset. Tracked points were visually inspected, and frames with low tracking confidence were excluded from downstream analyses. Due to the limitations of the dorsal view, fin-surface contact was estimated based on fin posture and motion; a fin was considered in contact with the substrate if it appeared flattened and stationary within the frame.

To assess the similarity of our model to the kinematics of four additional types of walking fish (sculpin, snakehead, lungfish, and catfish), we digitized their movements using publicly available video recordings[35–39]. Due to variability in video quality across publicly available recordings, automated tracking was not feasible for all datasets. In these cases, midline coordinates were manually digitized using CurveMapper, a custom MATLAB-based tool for extracting and fitting body midlines from video data (MATLAB R2024b; MathWorks, Natick, MA, USA). Digitized midline points were interpolated using a spline to generate 200 evenly spaced two-dimensional body coordinates per frame, parameterized from the tip of the snout(0) to the tip of the tail (200). Each video was rotated such that the fish moved consistently from right to left within the frame. We manually identified the starting and ending frames of individual gait cycles, where the tail of each fish was pointing upward in the frame (i.e., pointing to the right of the fish's body). We then applied the same processing pipeline used for *P. senegalus*, including downsampling, when necessary, to obtain between 10 and 13 midlines per cycle, and manual labeling of the midline and relevant anatomical landmarks. We did not consider cycles where the fish was at the extremities of the frame or when it did not walk straight, as turning behavior likely biases the kinematics.

To average the results of all the gait cycles for each fish, we linearly interpolated between the measured data points such that all gait cycles had one hundred equally spaced points between the start and the end of the gait cycle. There were two cycles from two videos of lungfish, nine cycles from six specimens of bichirs, twenty-two cycles from five videos of catfish, fourteen cycles from four videos of snakeheads, and one cycle from one video of a sculpin. For experiments of speed vs. frequency, the dataset was expanded to include two additional bichirs and one more video of a previous specimen for which the midline tracking was too noisy to collect joint angle data, but sufficient to track speed per tailbeat.

Kinematic data were normalized to body length to allow for direct comparison across species, removing the need for calibration to absolute spatial units. To calculate the walking speed of *P. senegalus*, we divided the Euclidean distance traveled by the body length of the specimen and the time over which it walked.

In this work, we rely on videos of walking fish available online and from previous work. We attempted to balance the need for a significant sample size of gait cycles with the suitability of the videos for analysis. As such, we discarded videos or cropped the length of videos where the fish is walking in a curved path, close to the extremities of the frame, or when an extreme fish-eye distortion was present. However, we note that one video of *P. annectens*[35] has some noticeable shake, and one video of *Hoplosternum sp*[36]. is noticeably not from an overhead view (angled view), and two videos of *C. argus*[38] were taken

using a fish-eye lens, though we cropped these videos to get gait cycles close to the center of the field of view.

## Discretization of the fish body into straight segments

To define the three straight segments based on the fish midline, we first calculated segment lengths according to the desired 2:4:5 ratio. We computed the cumulative distance along the midline from the snout (first point) to each subsequent point and identified the points at which this cumulative distance most closely matched the target lengths for the head and middle segments. The first segment extended from the snout to the point corresponding to 2/11 of the total body length. The second segment spanned from that point to the next point at 6/11 of the body length. The remaining portion of the midline defined the tail segment, terminating at the tip of the tail. We then calculated the joint angles between the head and middle segments ($\theta_2$), and between the middle and tail segments ($\theta_3$), with an angle of 0° defined as full alignment between adjoining segments. In addition, we quantified the forward translational speed and extracted the time-series heading of each fish to assess overall trajectory and body orientation throughout the gait cycle.

We used a least squares method to fit the tracked joint angles in the *P. senegalus* experiments to sinusoids of the form $\theta = A\sin(2\pi x + \phi) + C$, where $A$ is the amplitude, $x$ is the normalized fraction of the gait cycle, $\phi$ is the phase, and $C$ is a constant offset. Here, we fixed the frequency to $2\pi$ as we have defined each set of data as a complete cycle.

## Simulation design

Our physics-based simulation was created in MATLAB R2024b using the Simulink Multi-Body environment. We created simulated robots at two different size scales, one with a total body length of 0.1 m, similar to the size of the biological inspiration *P. senegalus*, and one with a total body length of 0.37 m, similar to the size of a physical robot. The geometries of the segments in the simulation of the fish-scale robot were based on measurements of *P. senegalus* specimens, while the geometries of the components in the larger simulated robot were based on the actual components of the physical robot discussed in the Methods subsection "Physical robot design and experiments". The rigid links were designed in CAD software, and the geometry was imported into the simulation environment to preserve the desired curved shapes.

The contact between the simulated robot and the ground was generated via a standardized model that approximates the contact as a smooth spring-damper in the normal direction and smooth stick-slip friction in an orthogonal direction (Supplementary Table 2). We did not prescribe the location of the contact points on the robot and the surface; these points were determined by the geometry and mass distribution of the robot combined with its posture and dynamic movement. We defined the motion of the joints via time-series angular position inputs.

The position input to the fin joint $\theta_1$ was a sinusoid that saturated at the maximum amplitude; the position input to the head joint $\theta_2$ and the tail joint $\theta_3$ was a full sinusoid. We assume that there is sufficient capacity for torque generation in each of the joints to satisfy the angular positions over time. For data collection, we defined the forward direction as collinear with the starting position of the middle link and positive in the direction of the head. We used the same method to calculate the walking speed of the simulated robot as we did with the walking speed of *P. senegalus*: we divided the Euclidean distance traveled by the body length of the specimen and the time over which it walked.

The head of the robot is attached to the world frame with a six-degree-of-freedom joint so that the robot can move freely in space, and its translation in the world frame is tracked for calculation of the walking speed. The joints are driven by angular position input, and torque is automatically calculated by the simulator based on the

defined geometry of the body elements and the inertia given by the defined mass. The ground is an infinite flat plane, and contact is evaluated between the plane and the rigid fin and body segments based on whether the geometry of the body element intersects with the plane. There is a normal force applied from the plane to the robot at the point of contact, defined as a spring force and damping pressing upward, opposing the intersection between the body element and the plane. There is also smooth stick-slip friction with static and dynamic elements opposing the motion of the element in contact with the surface. The position and orientation of the robot in the world frame are determined by the robot's interaction with the surface and with gravity, based on the posture of the robot, which is defined by the geometry of the rigid elements and the states of the joints. The simulation uses the ode45 simulator with a variable step size (maximum size 0.01 s). The full simulation code can be found online.

To calculate the Euclidean distance between the segment lengths (Fig. 8a), we use the following equation:

$$d = \sqrt{\left(s_{1,base} - s_{1,morph}\right)^2 + \left(s_{2,base} - s_{2,morph}\right)^2 + \left(s_{3,base} - s_{3,morph}\right)^2} \tag{1}$$

where $s_{n,base}$ is the length of the nth segment of the base morphology, and $s_{n,morph}$ is the length of the nth segment of the altered morphology.

### Physical robot design and experiments

We constructed a physical robot designed to replicate the morphology and motion constraints of the simulated undulating tripod model with parameters similar to the large-scale simulation (Supplementary Table 3). The robot consisted of three rigid segments, 3D printed in PLA plastic, connected by revolute joints actuated with servo motors (Robotis Dynamixel AX-12A). Because the maximum angular speed of these motors was about 1 rev/sec, we reduced the frequency of the physical experiments to 0.5 Hz so that we could still achieve constant frequency actuation even at the highest joint oscillation amplitudes. To ensure full autonomy during experiments, the robot carried its own power source housed within the central segment, and a microcontroller mounted on the tail segment, eliminating the need for external power or control tethers. The robot was actuated by transmitting pre-programmed angular position commands to each servomotor, corresponding to the desired joint trajectories derived from the simulated gait.

To measure locomotor performance, we recorded overhead videos of the robot's motion using a top-down camera. A visible marker was placed at the midpoint of the front edge of the robot to track its position. We used the open-source software Tracker (Physlets) to digitize the trajectory of the marker in two-dimensional space. Robot speed was normalized to body length per second (BL s⁻¹) by calibrating the spatial scale based on the known width of the robot's front segment. We used the same method to calculate the walking speed of the physical robot as we did with the walking speed of *P. senegalus* and the simulated robot: we divided the Euclidean distance traveled by the body length of the specimen and the time over which it walked.

### Reporting summary

Further information on research design is available in the Nature Portfolio Reporting Summary linked to this article.

## Data availability

The authors declare that the data supporting the findings of this study are available within the article and its Supplementary Information files. Source data are available as a source data file. Additionally, the publicly available videos used in this study that were not recorded by the authors are cited[35–39]. Source data are provided with this paper.

## Code availability

The code used for the analyses can be found via GitHub at https://github.com/mishidasan/undulating_tripod_gait_simulink_rev1.

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

## Acknowledgements

The authors thank the Di Santo, Iida, and Shubin lab members for interesting conversations on fish biomechanics and robots. This work was supported by a Research Grant from HFSP: RGP0010/2022: https://doi.org/10.52044/HFSP.RGP00102022.pc.gr.153619 received by V.D.S., N.H.S., and F.I.

## Author contributions

M.I., F.B., N.H.S., V.D.S. and F.I. conceived the work. M.I. developed the theoretical, simulated, and robotic models and designed and performed the experiments with the simulated and physical robots. F.B. and V.D.S. recorded the videos of *P. senegalus*. F.B. and T.P. tracked the kinematics of the various fish species from the videos. N.K.H. developed the data pipeline and performed the computer vision tracking for the initial biological analyses. M.I. performed the data analysis and wrote the manuscript, and all authors discussed the results and edited the manuscript.

## Competing interests

The authors declare no competing interests.
