## [Transparent Peer Review file · Nature Communications]

The undulating tripod gait as a model of the locomotion of walking fish

Corresponding Author: Dr Michael Ishida

Version 0:

Reviewer comments:

Reviewer #1

(Remarks to the Author)

In this manuscript the authors use a combination of live animal experiments, simulations, and robotic experiments to determine the minimum biomechanical necessities to facilitate walking in fishes. They begin by using data from five species to show that the tripod gait model of walking is present across a phylogenetically diverse group of fishes, thereby establishing the broad applicability of the study. They then focus on one species, *Polypterus senegalus*, to establish a biological basis for a motion pattern that further experiments can be compared against. Next they use simulations of a simplified system to test the effectiveness of this biological motion pattern compared to motion patterns not observed in animals. Finally, they use a physical robot to validate these simulations in a real world example. They then draw broad, but reasonable, conclusions about the implications of these results on the evolution of extant fishes and about the origins of terrestrial locomotion.

Overall, I think this study is a model of how impactful biomechanics experiments should be conducted in the modern age. The questions posed are deeply interesting and relevant to a broad swathe of biologists. Particularly the question of what the minimal requirements are to allow walking is relevant to the ecology of many clades as well as the evolution of terrestrial locomotion in tetrapods. Not only that, but this question is extremely well suited to the methods used, which means that the conclusions are convincing and satisfying. The study and the data are well presented and easy to understand, and it is well written throughout. I only have one major comment that needs addressing, and I've also provided some minor suggestions below that may help with clarity in a few places.

Major comment:

L268: I don't think it is fair to conclude that the robot moves fastest when the tail angle is 2.1 radians. In figure 6 this is the highest point, but it is still trending upwards so it seems believable that the speed could continue to increase. I know that the physical robot can't go beyond this angle, but as the authors state, the simulations are not limited by this constraint. I think this point would be much stronger if there was another data point demonstrating a higher tail angle. This is especially true because one of the conclusions is that *P. senegalus* is operating at a performance maxima, but this isn't fully supported without extending the simulation space beyond what was observed in the animal.

Minor comments:

L57: I don't really follow this logic. Although swimming and walking both involve complex use and coordination of the axial and appendicular skeleton, this doesn't necessarily mean that they are governed by shared principles. It seems like the mechanical demands will still be entirely different and the ways in which they coordinate their movement on each substrate could be unrelated. I might be misunderstanding what the authors are trying to convey here, so clarifying this or modifying the statement would help.

L59-60: It's interesting that bichirs and roach use different gaits given that they are extremely closely related. This could be worth mentioning to highlight the evolutionary lability of these behaviors. Especially given the fact that the authors claim that the tripod gait is convergent between clades, but without showing that it's a trait with variation it could just be an ancestral trait that is carried through most fishes.

L171: I think it's worth citing the phylogeny in Fig. 1 at the same time that Fig 4a is cited to make sure readers understand that this is the same phylogenetically diverse group presented in the first figure.

L204: I'm not sure where the ratio 2:4:5 comes from. I don't have any problem with it, but in the methods the authors state that it was the "desired ratio". I'm assuming that this was based on morphological measurements, but that should be clarified.

L407-408: I'm curious if the authors still think that the physical robot serves as a good confirmation of the simulations given that its performance didn't keep up with the simulations in most cases. This seems worth directly addressing in the results.

Figure 2: It seems like the top view should have dark blue points as well to indicate when there is ground contact on the pectoral fin.

Figure 5: It would be helpful to include the values of the constant parameters in the figure legend instead of just in the main text.

Figure 6: It would be helpful to include the values of the constant parameters in the figure legend instead of just in the main text.

Figure 8: What is "Morphology difference" on the x-axis in panel a? If this is a categorical variable then it should be more appropriately named and there shouldn't be numbers on the axis. Additionally, I think it would be helpful to have lines connecting points of the same color in panel b, as is done in the rest of the figures.

Figure 9: Again, it would be helpful to state the constant parameters in the figure legend, and to connect the points in panels d and e.

Supplemental videos: All the supplemental videos showing simulations would be much more clear at 0.5x or 0.25x speed. At their current speed it's difficult to really see what's happening.

(Remarks on code availability)

Reviewer #2

(Remarks to the Author)

The manuscript "The undulating tripod gait as a model of the locomotion of walking fish" by Ishida et al. presents a rigorous and intriguing investigation of their proposed "undulating tripod gait" model both in numerical/simulation space and in the real world with a robot. The model is a compelling and simple representation of the locomotion of *Polypterus senegalus* and likely has potential to describe general patterns across species, as argued by the authors, however, there are some challenges with the biological dataset makes it difficult to assess whether the described patterns are generalizable across species that use the "undulating tripod gait" (further detail provided in "Major Comments"). The discussion also lacks a solid grounding in literature (much of it is made up of paragraphs that lack any citations). I have placed some suggestions for how to improve this in the Minor Comments below.

MAJOR COMMENTS

While the biological dataset is not the focus of this paper, the authors rely on the idea that "consistent sinusoidal patterns" are seen across species at their segment joints and use this as a basis for the generalizability of their model. At present, this claim appears exaggerated because, if I have understood correctly, the authors have analyzed only one gait cycle for one individual from species other than *Polypterus senegalus* (and only one gait cycle for each of four individuals for *P. senegalus*). These small sample sizes make it difficult to ascertain whether the data represented are indeed indicative of consistent patterns, especially for the species other than *P. senegalus*. The range of speeds in the four *P. senegalus* individuals makes their case reasonably convincing, but an increased "sample size" all around (even if this is simply more gait cycles from each individual) will help support the assertion above. Further, it would allow presentation of an average pattern for each species, which likely better represents general patterns (e.g. are the two peaks in the lungfish trace for θ_2 commonly observed? Or a quirk of this particular gait cycle [or individual]? Or perhaps a consequence of their use of the head as a prop rather than their pectoral fins?).

To help improve the biological dataset the authors could: make use of additional gait cycles already captured (publicly available videos used to collect the current data often have multiple gait cycles), collect new recordings (if animals are available for experimentation), or procure videos from additional (publicly available) previous accounts of the terrestrial locomotion for these species (*Polypterus senegalus*: Standen et al., 2014; Standen et al., 2016; Foster et al., 2018 | *Protopterus annectens*: Horner and Jayne, 2014 | *Hoplosternum* sp.: Bressman et al., 2021).

The mechanism of movement also varies between these species. There are at least two notable differences that should be mentioned in the paper as they likely influence the nature of contacts between the animal and the substrate: in *Protopterus annectens* the "prop" is the anterior of the skull, and in *Hoplosternum* the pelvic fins (and mouth) also contact the ground and potentially constrain the contact points on the animal. These differences in locomotor structures likely impact the kinematics and contribute to differences in Figure 2 (and perhaps change the "optimal", or perhaps appropriate, segment lengths for

these different species).

Further, the substrate across which the animals are moving will impact the physical forces in the system and can affect the kind of movement the animal is capable of (e.g. Standen et al., 2016 and Bressman et al., 2019). This is beyond the scope of the paper to address in a modelling or robotic context, but it bears mentioning with regards to the biological datasets as the type of surface and physical properties of the surfaces that each species is moving across is different. For example, if the authors used movement of *Polypterus* on gravel as their exemplar (Standen et al., 2016), they would likely have reached a dramatically different conclusion.

Finally, some of the videos used for data collection suffer either from the angle of the camera (not directly above the fish) or from the type of lens (fish-eye lens), yet there is no mention of these challenges in the methods. Both of these video characteristics have the potential to influence and/or bias the kinematic (especially midline) data collected. Was any correction used for these characteristics? If so, what corrections were used? If not, the authors need to at least be transparent so that the reader can assess for oneself the level of confidence in the videos. That being said, there is at least one cycle of movement from a view directly above or directly below a fish in publicly available (even without contacting authors) videos between the references already used and those I have listed above. I urge the authors to use videos that will yield the most reliable kinematic information available, or to be transparent about the ways that the data may have bias.

MINOR COMMENTS

Lines 42-50: The authors have outlined a mechanism of generating movement that relies on “coordinated axial and appendicular movements”, however the first example they provide (mudskippers) are generally understood to use very little (if any) body movement to generate forward locomotion. It seems like the focus of this section of text is the use of ground reaction forces rather than hydrodynamic forces during walking and the diversity of walking strategies, so I recommend rewording the first sentence (“On land, walking fish generate...”) to include all possibilities listed in the subsequent examples (e.g. “On land, walking fishes generate forward motion through axial movements, appendicular movements, or the combination thereof...”). This would avoid any incorrect implications in the examples described.

Lines 54-62: Here I find it somewhat confusing that the authors assert that “...fishes similarly blend axial bending with appendage-based support in ways that defy simple classification...” but then provide examples of locomotion that fit rather neatly into those classifications without any further explanation of their proposed “continuum”. Current consensus (e.g. reviewed in Pace and Gibb 2014 and Lutek et al. 2022) in the literature is that there are indeed species that use axial movements (e.g. ropefish and eels; perhaps primarily, though it is unclear if the fins provide additional propulsion in these cases), appendicular movements (e.g. mudskippers; perhaps primarily, but contributions of the body are unclear), and many that use a combination of axial and appendicular movements (the focus of this paper). I agree that there is likely extensive variation both within and between species for the axial-appendicular “mode” of locomotion in terms of the “contribution” of appendages and body (and perhaps there is more variation than has been acknowledged in the literature for the axial and appendicular “modes” as well), but whether this is indeed a “blending” that leads to continuity is still an open question, rather than a demonstrated characteristic of terrestrial locomotion in fish (and indeed, the authors really provide no evidence to support their claim of “continuity” in a terrestrial environment – the examples they use fit quite well into three distinct categories). I would suggest that this framing of terrestrial locomotion either needs some clarification, including references to support the assertion of “continuity” in the terrestrial environment, or I would suggest that the authors instead focus on the unifying “physical principles” in a terrestrial environment – these principles are indeed, as the authors point out, understudied and apply across the listed “modes” (and in both aquatic and terrestrial environments) regardless of whether the locomotion is a continuum or discrete categories. For example, the following version of this excerpt would shift the focus away from the continuum assertion (which really isn’t the focus of this paper anyway): “Rather than falling into discrete kinematic categories, swimming behaviors form a continuum shaped by the interplay of body form, flexibility, and flow characteristics [10–12]. Similar interplay is likely involved on land as walking fishes can use both axial bending and appendage-based support which may be governed by shared, substrate-independent mechanical principles. Some species have been described to primarily articulate their pectoral fins (e.g., mudskippers [13]), some to primarily articulate their vertebral columns (e.g., ropefish [14]), and some to use a combination of both (e.g., bichirs [15]). However, these categorizations use kinematic observations to classify the gaits of different species but do not explore the underlying physics principles of the gait.”

Lines 90-94: The figure legend should list the species for which the snapshots are provided, where these snapshots are from, and how the snapshots were selected.

Lines 107-112: The figure legend should include some description of where these snapshots are taken from (I assume the top panel is spaced evenly throughout a locomotor cycle, but I’m not sure when the bottom panel snapshots represent).

Lines 115-122: Perhaps a reference to Figure 1 would be helpful here since you have nicely illustrated this behaviour here.

Lines 178-179: Because of the large variation in the range of theta in Figure 4b and 4c it might help the reader focus on patterns of angle fluctuation (“sinusoidal patterns”) to normalize the traces for each species to their maximum and minimum (such that values range from -1 to 1). Especially since the authors are focussing on phase relationships, this would help readers visualize the relative timing of maximum and minimum angles. If the biological dataset is augmented (as I suggest above) then I would also recommend directly presenting the phase relationship between θ_2 and θ_3 . While I agree in principle that a conserved phase relationship is evidence that all of these walking gaits could be described by a single model, the data as currently presented does not sufficiently support this assertion.

Lines 184-193: At present it is unclear the sample sizes for the line plotted in Figure 4b,c and e. Please specify whether the data is an average or from a single gait cycle.

Lines 192-193: Please clarify whether the “anterior contact point” in the gait diagram (Figure 4e) is the “prop” or the anterior body contact.

Lines 247-249: Can the authors provide either video evidence or quantification of this such that readers can assess this claim?

Lines 264-267: I wonder if the *P. senegalus* data points would be more useful if plotted at the specific amplitude for that gait cycle, rather than at the average θ_2 and θ_3 amplitude. Presumably a sine curve could be fit to each analyzed gait cycle (or the amplitude could be determined empirically) and that this would provide a more precise comparison between these values for the fish and those for the simulation.

Lines 304-306: The panels (a) and (b) in this figure are not correctly identified in the figure caption. Additionally, as for Figure 6, I believe it would be more appropriate to plot the *Polypterus* values at the measured phase for each gait cycle.

Lines 307-326: Including additional figure references throughout this section where appropriate (and perhaps including a figure that depicts the relationship between speed and ϕ_3) will help clarify the statements here.

Lines 335-350: Include figure references throughout these paragraphs to direct the reader to where this data can be found.

Lines 411-415: Unclear if a “full factorial” set of combinations (i.e. all possible combinations) was used for these tests, or if a subset were tested. It is therefore also unclear what combination of surface and fin frictions is presented in Figure 9f.

Line 433: I would caution against the use of the word “efficiency” here – the authors have not actually measured any values that quantify efficiency (e.g. cost of transport) but rather have quantified the speed of movement. Perhaps an argument could be made for “peak locomotor performance”, although “performance” is also a loaded term that usually requires measures beyond just speed. Perhaps the most precise and appropriate phrasing would be “peak locomotor speed”.

Line 436: Perhaps I've misunderstood, but I believe the results presented also suggest that frequency also modulates locomotor speed (Figure 5).

Line 439: I believe this should read “... morphology of *P. senegalus*, aligned closely with...”

Lines 449-451: While not perhaps exactly the same as the walking discussed here, the idea that “many fishes may possess latent terrestrial capabilities” has been investigated for jumping in several species of fish and would perhaps be worth a mention and/or citation here. See as a starting point Gibb et al., 2011 and Gibb et al., 2013. This would also help make the discussion more rich and well grounded in the literature.

Line 458: It is unclear to what the authors are comparing the magnitude of muscular force here. For the few species that have been studied, terrestrial locomotion tends to require higher magnitude muscle activation (which presumably produces higher forces) than those used for aquatic locomotion (e.g. Eels, *Polypterus*, *Rivulus*, Lungfish; reviewed in Lutek et al., 2022). If the authors are comparing this theoretical muscle output to what would otherwise be required for terrestrial locomotion, this should be made explicit.

Lines 458-459: “...specialized stepping gaits that involve active body lifting...” See, however, Standen et al., 2014 - *Polypterus* raised on land do indeed lift their bodies (as measured by nose elevation) higher off the ground than fish raised in water, suggesting that at least for the focal species of the present manuscript lifting of the body off the ground is indeed an important part of this kind of “gait” on some substrates. It would also help strengthen this statement to provide examples (and appropriate references) for the “specialized” gaits to which the authors are referring.

Lines 460-462: The statement “This may be particularly advantageous in...” is absolutely true in theory, but at present there is little evidence (or perhaps “conflicting” evidence) for whether walking (or for that matter terrestrial locomotion as a whole) in fish is anaerobic. Many amphibious fish species possess specializations that enable respiration on land. In fact all of the example species used in this paper have specializations: *Polypterus* have a lung, lungfish have a lung, armoured catfish have an accessory respiratory organ, snakeheads have a suprabranchial organ, and tidepool sculpins can perform cutaneous respiration, and there are various accounts that support both the use of anaerobic (e.g. eels white muscle activity tends to increase on land; Gillis 2000, *Polypterus* pectoral fin fiber type switches to “fast” fibers; Du & Standen, 2017, Mangrove *rivulus* have higher “body muscle” activity on land; Perlman and Ashley-Ross, 2016) and aerobic (e.g. red body muscle activity tends to be higher on land than in water for *Polypterus*; Foster et al., 2018, Mangrove *rivulus* increase the size of red, but not white muscle fibers after terrestrial acclimation; McFarlane et al., 2019) muscles for terrestrial locomotion. This makes it difficult to assert with any certainty that the systems investigated are using anaerobic mechanisms to move. This is indeed an interesting possibility and would benefit from a more complete discussion of this idea and the implications of the author’s model when interpreted in the context of the whole of the terrestrial fish locomotion literature.

Lines 504-591: Many of these paragraphs are strikingly lacking any grounding in the present literature. While I am more well versed in the literature of the biology of amphibious locomotion than robotics and models, I wonder if there are places where

relevant literature would help strengthen the discussion. For example, lines 524-526 discuss the impact of morphology and friction on terrestrial locomotion. I believe this would have been published after this paper was submitted, but Lopez-Chilel and Bressman (2025) have addressed the impact of mucus and scale anisotropy for terrestrial locomotion in fish which seems directly relevant here. Likewise, lines 529-532 would benefit from explicit discussion of (and reference to) the relevant examples mentioned indirectly.

Lines 594-615: Be sure to explicitly mention the sample sizes for each species (number of individuals and number of cycles analyzed).

movieS1 & S4: It would be helpful to have a scale bar to assess speed of the animal, if possible.

(Remarks on code availability)

I was indeed able to run the code. The README is sufficient, but some further explanation of how to change parameters (or examples of what those parameters should be set at to get the different situations presented in the paper) would be helpful.

Reviewer #3

(Remarks to the Author)

The authors present a study of the locomotion adaptations of walking fishes that engage in the “undulating tripod gait.” They employ simulations, biological observations, and robotic proxy experiments to investigate the impact of gait and morphological parameters on locomotion. The metric of interest, in this case, is forward speed of the animal/robot. The authors systematically vary different parameters, such as the amplitude or phase offset of sinusoidal inputs to the simulated robot’s joints, and measure the resulting forward speed. They compare results calculated, when relevant, with those gleaned from observing the biological organism. Ultimately, the authors state that the simulation/robotic proxies recapitulate biological observations to a reasonable degree. From these, they make remarks about the undulating tripod gait as a simplistic, widely-seen locomotion strategy, and hypothesize about its utility in the transition from aquatic to terrestrial locomotor modes.

The science presented herein is interesting, and the experiments provide a strong foundation for a compelling story. However, a number of issues with the manuscript collectively indicate that it is not yet ready for publication. These issues include: unclear figures, sometimes unclear documentation of methods, insufficiently justified choices for parameter sweeps (prompting many whys? throughout), insufficient amount of biological experimental data (making for very sparse comparisons with experiments and difficult to see trends), insufficient amount of simulation model experimental data, and unconcise/redundant discussions. Among major points are the following:

-Simulation has the ability to sweep massively through the parameter space with virtually no additional expense. Why not look at results for more of these preternatural parameters, pushing to the limits of what would be possible in physical systems? Instead of generating sparse data points with simulation, you have the opportunity to generate “surfaces” that give a much more continuous intuition of how parameters interact in your experiments.

-There does not appear to be a lot of cohesion between the parameters evaluated in the simulations, the actual animal, and the robot. More data would seem necessary so plots are not, for example, 10 simulation points and 3 biological points. It would be compelling if (for the cases possible to evaluate on the animal and robot), a large set of parameters was plotted together to show more comprehensively agreements / disagreements between sim/bio/robot.

-You make interesting observations that the animal seems to fall into some sort of range consistent with the best performing simulations. Is this “locally optimal” parameter range for forward locomotion speed maximization representative? I’m not sure, due to the limited number of experimental datapoints and from the fact that simulation did not cover a broader range in the parameter sweep. Furthermore, are there certain conditions, e.g. fight or flight, wherein the animal would exhibit a faster gait? Do these types of situations, perhaps, when observed in the wild, influence the undulating tripod gait ‘basis,’ deviating it from the nominal case that you studied in the lab?

-A large portion of the discussion section is repeating previous content and long-winded. Better to opt for a more concise description of the takeaways of the present study, rather than a retrospective justification of the methodologies and a rehashing of the take-aways nearly verbatim from prior sections. Everything from “Our findings reinforce the value of minimalistic models and robotic analogs in uncovering unifying principles of locomotion.” Especially applies to this comment.

Consider other points below, which occur in roughly the same order they can be found in the manuscript.

-The term “axial,” as in “axial body undulation” does not give a concrete image, and should be better described from its first mention. “Axial” is also used in the abstract and leaves the reader wondering which “axis” this motion truly lies about. It would benefit from an early illustration, e.g. in figure 1.

-“Despite this shared reliance on substrate interaction, species differ markedly in their terrestrial walking strategies.” Seems to contradict abstract statement about remarkably similar gaits.

-“Despite striking morphological differences, many fishes exhibit convergent patterns of undulatory locomotion. Rather than falling into discrete kinematic categories, swimming behaviors form a continuum...” Again, seems to contradict/confuse a bit with the first statements.

-The entire bit from “Researchers use animal-inspired models to investigate the principles underlying specific biological phenomena...nature.” is fine justification, but is quite generic and long-winded. I’m not sure it helps funnel the scope of the

paper, when a succinct justification for the use of robot template models is already given in the previous sentence.

- The term “convergent gait” should be defined for the wide readership of nature communications.
- “ The undulating tripod gait is different from other fish gaits that neglect...” How, exactly, is it different? Would help to be more specific here.
- Fig. 2 would be more clear if roll/pitch/yaw axes were formally defined with some coordinate frame.
- “ When the body is propped on the left side of the midline, the fish swings around the pivot point in the counterclockwise direction and vice versa when the body is propped on the right side of the midline.” Are you referring to Fig 2 here? If so, a reference would help.
- “In this model (Fig. 2), we represent the propping motion as being generated by a rigid beam rotating around the fish’s head-tail axis (θ_1) such that $\theta_1 > 0...$ ” For easier interpretation, please label these variables directly on the figure.
- “Locomotion results from the interplay of gravitational forces, joint trajectories, and substrate interactions.” Self-evident statement; consider removing.
- “In this work, we focus specifically on the terrestrial locomotion of *Polypterus senegalus* as an exemplar of the undulating tripod gait, in which the fish uses its pectoral fins as propping appendages.” Why this particular animal? Why is it a good representative of the gait, out of the others mentioned?
- “We tracked the midlines of five different species that exhibit the undulating tripod gait: *P. senegalus*, the African lungfish *Protopterus annectens* [34], the armored catfish *Hoplosternum* sp. [35]...” Here, and this also applies to all other “methods,” it would facilitate ease of reading / understanding if a specific corresponding section in the methods was referenced.
- Fig. 3 should have a scale bar indicating the size of the organism
- “ Across species, the fish alternates which side of the body makes contact with the substrate and subsequently sweeps the tail toward that planted side (i.e., after engaging the left pectoral fin, the tail bends leftward).” Point of clarification - if this is saying something to the effect of: “all observed species showed this behavior” (which I think if your intention), it should read “alternate,” since fish is plural. On the other hand, if it’s a single fish type that exhibits this behavior, then you should specify which one.
- Unclear terminology: “angular time series”
- “ representative exemplar.” Redundant wordage
- Fig 4b-c: what’s the sampling rate? It seems to result in fairly jagged curves. Can you be sure you’re capturing the true behavior if the sampling rate is so low?
- “ We used a generic contact model consisting of a spring and damper...” Would like to see more information and explanation about the simulation setup, perhaps in the SI/methods, to help with reproducibility. I am glad to see the code is included in the submission.
- “ We approximated the behavior of the fin joint as a saturating sinusoid, where the joint reaches a threshold position and holds a constant angular position analogous to the fish holding its fin steady against the ground.” Not clear what “holding its fin steady against the ground” means, and how it relates to the locomotion; what is the connection to cyclic motions with no “holding of configurations” involved?
- Fig. 5 does not appear to have sublabels.
- “ The measured speed of *P. senegalus* was slower than that of the simulated robot, although the discrepancy between the speed of the simulated robot and the real fish was small at higher frequencies (Fig. 5b).” Where is this difference plotted?
- How are representative geometries for the simulations defined? The radius of the cylinders, for example --- these change the inertia of the appendage, as well as the possible contact locations when walking. These, in turn, influence the resulting forward velocity.
- For statements like “The simulated robot moves fastest with the amplitudes of $\theta_2 = 1.5$ and $\theta_3 = 2.1$, which are the same gait parameters observed in our experiments with *P. senegalus* specimens.” It would be nice to see a hypothesis, grounded in physical explanations, as to why.
- Statements like “significantly affects the forward speed of the robot” are qualitative in nature, and would be better understood with numbers and some sort of statistical analysis. For instance, what is the ‘sensitivity’ of the parameter vs that of θ_3 --- which one has more bearing, quantitatively, on the performance? Perhaps PCA would be useful here.
- Are quantities like θ_2 truly constant for the real animal throughout its gait cycle?
- “ When the phase of the head joint is zero ($\phi_2 = 0$), the head segment begins in-line with the middle segment and begins by rotating clockwise with respect to the joint (i.e., the head rotates to the right of the body). When the...” I think the descriptions of phase should be supplemented by a simple illustration, to help readers grasp their definitions more quickly.
- Fig. 7: are the captions switched? Does not make sense otherwise.
- “For this set of simulations, we reduced the frequency of the gait to 0.5 Hz because of the larger body size with more inertia.” More information on why would be nice – is this just the physical limit of the hardware you tested on? Are you trying to keep another physical parameter constant?
- “ discretizing” -> discretizing?
- “ The maximum speed of the simulated robot in either of these configurations with two body segments was 5% of the maximum speed with three body segments.” Would like to see more information about these experiments, with comprehensive results, in an SI.
- “The first variation is with all three segments of equal length (1:1:1 ratio of the segment lengths).” Would be clearer to specify that this is with respect to the original. I know it’s implied, but upon first readthrough, I did each segment length was just equivalent.
- Fig. 8: Would be interesting to see more supernatural embodiments, since the simulation is “cheap” to run anyway. For example, why not 1:1:9? On another note, the x-axis of a) does not make sense to me – I don’t see how the ratios map to scalar values on the x-axis.
- “... this experiment implies that the same gait will generate locomotion of different speeds when applied to different body morphologies.” This is not particularly surprising – is this would be curious to see more elaboration, especially discussion of the supernatural body types.
- Fig. 9d-e: undefined error bars and sample sizes

-Best to cite coefficients of friction rather than just saying 'low' and 'high'
-From the video, it looks like the experimental robot struggles to move straight – do you change calculation of the velocity for these deviating trials?
-“ Unlike traditional kinematic analyses, which only observe how animals move, physics-based robotic models can test how and why certain motions lead to successful locomotion under well-defined constraints [17]. In particular...” Seems like a strange place to justify the methodology, in the discussion. Also seems redundant.

(Remarks on code availability)

A little more organization for the repo would help with reproducibility; why are there "robot" and "fish" directories with the same readme.tex in each?

Version 1:

Reviewer comments:

Reviewer #1

(Remarks to the Author)

I appreciate the author's work to address my comments. The changes (or justifications in a few of the minor comments) are thorough and I no longer have any major comments. There are a few minor points that the authors should address listed below, but they are easily addressed and don't change anything substantive about the paper. The manuscript as it now stands is convincing of the importance of the undulating tripod gait in the evolution of terrestrial locomotion in fishes. It does an excellent job of dissecting the behavior through multiple evidentiary avenues, and is an excellent step towards understanding the biomechanical basis that is necessary for this locomotor style as well as how variation in morphology and kinematics affect it.

Minor comments:

Abstract:

"In this work, we model of the undulating tripod gait" should be "...we model the undulating..."

Intro:

The first two paragraphs are a bit confusing to read. The narrative switches between terrestrial and aquatic locomotion several times, while also switching between focusing on convergent and divergent morphology/kinematics.

Methods:

I'm aware that other reviewers asked for additional details in the methods, which is why the authors expanded these sections. While I agree that this additional information is important to make available, I'm not sure that it should all be in the main text. I think the previous version felt more succinct and better fit the format of the journal. As such, I would gently suggest moving the following to a supplemental text file:

- Specifics of the DeepLabCut methodology
- Specifics of the midline tracking methodology
- Explanation of the walking speed measurement for the simulation and robot
- Formula for the Euclidean distance between segment lengths used in Fig. 8a

Figures:

Figure 8. I appreciate that the authors made the x-axis of 8a more quantitative. However, I don't think the new axis label is informatively titled. "Euclidean distance" is the method, but doesn't tell you what the meaning of the measurement is. Probably better to label it "total difference from base morphology (mm)" or something like that.

(Remarks on code availability)

I don't have simulink, so I was not able to run the code. However, the documentation for the code seemed appropriate.

Reviewer #2

(Remarks to the Author)

I appreciate the efforts that the authors have gone through to address the comments of myself and the other reviewers and I'm satisfied that their revisions sufficiently address said comments.

(Remarks on code availability)

Reviewer #3

(Remarks to the Author)

The authors addressed my comments quite thoroughly. I am happy to see major improvements in the quality of the manuscript. In particular, having resolved text ambiguities, collected additional data points for experimental plots, improved methodological details for reproducibility, and explicitly acknowledged limitations of the present analysis, the paper stands as much stronger. It is my recommendation to proceed with acceptance.

I have two minor comments.

-I think Fig 2 could benefit from coordinate systems, such that it is absolutely clear about the orientation of the “top” and “front” fish views relative to one another.

-grammatical subtlety in the abstract: “In this work, we model of the undulating tripod gait by approximating the fish’s axial undulation...” -> I wouldn’t use “the” because you have yet to specify what type of fish. Might be best to say you model “a fish’s” or “fishes’ “ and then agree the subsequent verb endings.

(Remarks on code availability)

The authors fixed the previous issue with the same README file in each directory; the code structure and instructions now appear just fine.

The images or other third party material in this Peer Review File are included in the article’s Creative Commons license, unless indicated otherwise in a credit line to the material. If material is not included in the article’s Creative Commons license and your intended use is not permitted by statutory regulation or exceeds the permitted use, you will need to obtain permission directly from the copyright holder.

We thank the reviewers for their comments and suggestions and believe that with their help, we have improved both the content and the presentation of the manuscript. The major changes we have incorporated in this revised version includes:

- 1) The inclusion of additional analysis of fish videos, some publicly available from previous work and some new experiments from our own *P. senegalus* specimens.
- 2) A few additional simulated robot cases and refined plotting of the existing data to better compare with the *P. senegalus* data.
- 3) Concision of the discussion, additional references, and clarifying text about the model and the conclusions we are making herein.
- 4) More thorough description of the methods.

Reviewer #1 (Remarks to the Author):

In this manuscript the authors use a combination of live animal experiments, simulations, and robotic experiments to determine the minimum biomechanical necessities to facilitate walking in fishes. They begin by using data from five species to show that the tripod gait model of walking is present across a phylogenetically diverse group of fishes, thereby establishing the broad applicability of the study. They then focus on one species, *Polypterus senegalus*, to establish a biological basis for a motion pattern that further experiments can be compared against. Next they use simulations of a simplified system to test the effectiveness of this biological motion pattern compared to motion patterns not observed in animals. Finally, they use a physical robot to validate these simulations in a real world example. They then draw broad, but reasonable, conclusions about the implications of these results on the evolution of extant fishes and about the origins of terrestrial locomotion.

Overall, I think this study is a model of how impactful biomechanics experiments should be conducted in the modern age. The questions posed are deeply interesting and relevant to a broad swathe of biologists. Particularly the question of what the minimal requirements are to allow walking is relevant to the ecology of many clades as well as the evolution of terrestrial locomotion in tetrapods. Not only that, but this question is extremely well suited to the methods used, which means that the conclusions are convincing and satisfying. The study and the data are well presented and easy to understand, and it is well written throughout. I only have one major comment that needs addressing, and I've also provided some minor suggestions below that may help with clarity in a few places.

Major comment:

L268: I don't think it is fair to conclude that the robot moves fastest when the tail angle is 2.1 radians. In figure 6 this is the highest point, but it is still trending upwards so it seems believable that the speed could continue to increase. I know that the physical robot can't go beyond this angle, but as the authors state, the simulations are not limited

by this constraint. I think this point would be much stronger if there was another data point demonstrating a higher tail angle. This is especially true because one of the conclusions is that *P. senegalus* is operating at a performance maxima, but this isn't fully supported without extending the simulation space beyond what was observed in the animal.

We thank the reviewer for this comment and we have performed this set of simulations and included them in Fig. 6b:

From this we see that $A_3 = 2.1$ does result in faster speed than $A_3 = 2.5$, though in some cases it is only a small improvement.

We have edited the text describing this result accordingly in Section 2.4.2 (page 11, paragraph 1):

The robot's speed also increases as the amplitude of the tail joint increases up to the maximum physically achievable angle $\theta_3 = 2.1$ rad and then reduces with further increase of amplitude achievable in simulation only (Fig. 5b).

And further down in the same section (page 12, paragraph 1):

With three rigid body segments, the tail segment contacts the ground and creates most of the pushing force, so the oscillation of the tail joint θ_3 affects both the magnitude and direction of the ground reaction force. As the amplitude of the tail

joint increases, both the range of motion and the relative velocity of the tail segment increase because of the larger angular sweep at the same frequency (Supplemental Movie S3). However, at the highest values of tail amplitude, the angle of the tail drives forces that are more horizontal than forward, reducing the effectiveness of the additional tail sweep.

Minor comments:

L57: I don't really follow this logic. Although swimming and walking both involve complex use and coordination of the axial and appendicular skeleton, this doesn't necessarily mean that they are governed by shared principles. It seems like the mechanical demands will still be entirely different and the ways in which they coordinate their movement on each substrate could be unrelated. I might be misunderstanding what the authors are trying to convey here, so clarifying this or modifying the statement would help.

We thank the reviewer for this point of clarification and agree that the mechanical demands will be different between the aquatic and terrestrial environments. This statement was intended to express the idea that there are shared mechanical principles at play in both environments that are tied to morphology. A fish with a specific morphology has a limited variety of motions it can make in either environment because of the constraints defined by its body. Here, we discuss previously developed categorizations of walking locomotion defined by axial articulation, appendicular articulation, and the combination of both and posit that our model presented in this work will provide additional specificity not possible within the previous categorization framework.

As such, we have rewritten most of this paragraph (page 2, paragraph 2):

With regards to terrestrial locomotion, previous work has instead attempted to classify walking fish using discrete categories such as using axial-bending, appendage-based support, or an axial-appendicular combination [7]. Of these, a few species employ almost entirely appendage articulation (e.g., mudskippers [13]) or almost entirely axial motion (e.g., ropecod fish [14]), but the vast majority of fishes exhibit both to a significant degree. It is difficult to further divide this main group into meaningful categories because of the extensive variation in locomotory kinematics between species. Species also exhibit different usages of their vertebral columns and pectoral fins depending on the environment in which they are moving [14,15], introducing further combination of these characteristic motions. Classifying the gaits of different species in this manner using these axial-appendicular categorizations relies on kinematic observations without exploring the underlying physics principles of the gait.

L59-60: It's interesting that bichirs and ropefish use different gaits given that they are extremely closely related. This could be worth mentioning to highlight the evolutionary lability of these behaviors. Especially given the fact that the authors claim that the tripod gait is convergent between clades, but without showing that it's a trait with variation it could just be an ancestral trait that is carried through most fishes.

We thank the reviewer for this comment. Previous work has described the ropefish as using slow, large-amplitude body waves without significant use of the pectoral fins (Pace & Gibb 2011) for terrestrial locomotion and notably do not describe this as "walking".

We have added this idea to the Discussion section (page 17, paragraph 2):

The widespread presence of the undulating tripod gait among distantly related taxa of walking fishes with varying but overlapping morphologies highlights the extent of locomotor convergence in terrestrial fish locomotion [40,41]. This convergence is not likely due to shared ancestry, as evidenced by the different terrestrial walking strategies of closely related species such as *P. senegalus* and *E. calabaricus* (both of the family Polypteridae), but rather due to common mechanical constraints imposed by body-substrate interactions.

L171: I think it's worth citing the phylogeny in Fig. 1 at the same time that Fig 4a is cited to make sure readers understand that this is the same phylogenetically diverse group presented in the first figure.

We thank the reviewer for the suggestion and agree - we have edited the relevant sentence in Section 2.2 (page 7, paragraph 2):

We found that this general gait pattern describes the walking locomotion of a diversity of fish species with different morphologies including those we highlighted in Fig. 1.

L204: I'm not sure where the ratio 2:4:5 comes from. I don't have any problem with it, but in the methods the authors state that it was the "desired ratio". I'm assuming that this was based on morphological measurements, but that should be clarified.

We thank the reviewer for pointing out this omission. This ratio for discretization comes from observations of the *P. senegalus* specimens. The "head segment" of length $L_1 = 2$ ends approximately at the pectoral girdle as there is almost no bending in the region more anterior than the pectoral girdle. The "body segment" of length $L_2 = 4$ qualitatively balances capturing bending in the middle of the body with bending along the tail.

We have edited some text to the section describing the tracking of *P. senegalus* to where the 2:4:5 ratio is introduced to make this clear (page 6, paragraph 1):

Based on observations of the walking motion, the lengths of these segments followed a 2:4:5 ratio, corresponding to 0.182, 0.367, and 0.455 of the body length (BL). With this ratio, the first joint was positioned near the pectoral girdle as we observed little bending to the anterior of that point and the second joint was placed slightly ahead of the pelvic girdle to capture both bending in the body and in the tail.

L407-408: I'm curious if the authors still think that the physical robot serves as a good confirmation of the simulations given that its performance didn't keep up with the simulations in most cases. This seems worth directly addressing in the results.

We thank the reviewers for this comment. We think that the physical robot is still valuable to ensure that the trends are correct and to show a proof of concept that this model bears out in the real world. There are a number of reasons for the discrepancy between the simulated and experimental robot that we generally attribute to the modeling of the surface contact. Modeling friction as a normal spring and damper and a simple linear relation for static and kinetic friction is not the most accurate way to model realistic contact.

We have made a comment about the need to bridge the simulation-reality gap in the Discussion section where we discuss the contact conditions (page 19, paragraph 1):

Notably, our robot's locomotion required a balance between high enough friction for reacting forces and low enough resistance to permit anterior gliding. We achieved these conditions both under simplified substrate mechanics in simulation and in the physical world, though more work should be done to bridge these two environments.

In addition, we have commented about further refining our understanding of the effects of contact (page 19, paragraph 2):

It would also be valuable to integrate models of energy cost, substrate interaction, and environmental feedback to assess gait efficiency and robustness under ecologically relevant conditions, while explicitly accounting for real-world surface properties such as compliance, viscoelasticity, and fluid layering.

Figure 2: It seems like the top view should have dark blue points as well to indicate when there is ground contact on the pectoral fin.

We thank the reviewer for this suggestion. The reason these points are not labeled in the top view is that while the sequence of the body bending and pectoral fin contacts are consistent across the species exhibiting the undulating tripod gait, the exact coordination between the bending motion and the contacts can vary. In Fig. 2 we

illustrate the contact for an example front-view case but are trying to keep the top view more general.

Figure 5: It would be helpful to include the values of the constant parameters in the figure legend instead of just in the main text.

We thank the reviewer for the comment and have added the following to the Fig. 5 caption:

In all cases, the other gait parameters were held constant and matched those tracked from *P. senegalus*: $A_2 = 1.5$, $A_3 = 2.1$, $\phi_2 = \pi/2$, $\phi_3 = \pi$.

Figure 6: It would be helpful to include the values of the constant parameters in the figure legend instead of just in the main text.

We thank the reviewer for the comment and have added the following to the Fig. 6 caption:

In all cases, we held the other gait parameters constant matching those tracked from *P. senegalus*: $\phi_2 = \pi/2$, $\phi_3 = \pi$. When varying the amplitude of θ_2 , we held $A_3 = 2.1$; when varying the amplitude of θ_3 , we held $A_2 = 1.5$.

Figure 8: What is "Morphology difference" on the x-axis in panel a? If this is a categorical variable then it should be more appropriately named and there shouldn't be numbers on the axis. Additionally, I think it would be helpful to have lines connecting points of the same color in panel b, as is done in the rest of the figures.

We thank the reviewer for this comment. In this original version of the figure, the "morphology difference" was probably best described as a categorical variable. We have instead updated Fig. 8a to replace the x-axis with a better quantification: the difference between the segment lengths of the new morphologies and the base morphology quantified by Euclidean distance.

The updated Fig. 8 caption is as follows:

a) Speed of simulated robot with the same total body length but different body proportions. Blue denotes the base morphology (2:4:5 ratio of segment lengths), red denotes the morphology with equal segment lengths (1:1:1 ratio of segment lengths), and yellow denotes the morphologies with two short segments and one long segment (either 2:2:7, 2:7:2, or 7:2:2 ratios of segment lengths). b) Speed of simulated robot versus amplitude of the actuated fin segment for different lengths of the fin segment L_0 . Each version of the simulated robot was tested with five amplitudes of θ_1 ; points not included in the plot were cases in which the robot could not produce stable locomotion and rolled.

To describe the new x-axis, we have added a line in the text (page 14, paragraph 2):

The walking speed of the simulated robot with these morphologies is plotted against the difference (Euclidean distance) between the new segment lengths and the base morphology.

We also have added a description of the calculation of Euclidean distance to the Methods Section 4.3 (page 22, paragraph 4):

To calculate the Euclidean distance between the segment lengths (Fig. 8a), we use the following equation:

$$d = \sqrt{(s_{1,base} - s_{1,morph})^2 + (s_{2,base} - s_{2,morph})^2 + (s_{3,base} - s_{3,morph})^2}$$

where $s_{n,base}$ is the length of the n th segment of the base morphology and $s_{n,morph}$ is the length of the n th segment of the altered morphology.

Figure 9: Again, it would be helpful to state the constant parameters in the figure legend, and to connect the points in panels d and e.

We thank the reviewer for the comment and have changed the relevant elements of the Fig. 9 caption:

d) Comparison of the speed of the physical robot and the simulated robot varying the amplitude of the head joint (θ_2) oscillation. We held the other gait parameters constant matching those tracked from *P. senegalus*: $\phi_2 = \pi/2$, $\phi_3 = \pi$, and $A_3 = 2.1$. e) Comparison of the speed of the physical robot and the simulated robot varying the amplitude of the tail joint (θ_3) oscillation. We held the other gait parameters constant matching those tracked from *P. senegalus*: $\phi_2 = \pi/2$, $\phi_3 = \pi$, and $A_2 = 1.5$.

The points in panels d and e are not connected because the plots get a little messy when lines are added to the points with small error bars.

Supplemental videos: All the supplemental videos showing simulations would be much more clear at 0.5x or 0.25x speed. At their current speed it's difficult to really see what's happening.

We thank the reviewer for this comment and have added the slow motion view of the simulated fish-scale robot in Supplemental Movies S2, S3, and S4.

Reviewer #2 (Remarks to the Author):

The manuscript “The undulating tripod gait as a model of the locomotion of walking fish” by Ishida et al. presents a rigorous and intriguing investigation of their proposed “undulating tripod gait” model both in numerical/simulation space and in the real world with a robot. The model is a compelling and simple representation of the locomotion of *Polypterus senegalus* and likely has potential to describe general patterns across species, as argued by the authors argue, however, there are some challenges with the biological dataset makes it difficult to assess whether the described patterns are generalizable across species that use the “undulating tripod gait” (further detail provided in “Major Comments”). The discussion also lacks a solid grounding in literature (much of it is made up of paragraphs that lack any citations). I have placed some suggestions for how to improve this in the Minor Comments below.

MAJOR COMMENTS

While the biological dataset is not the focus of this paper, the authors rely on the idea that “consistent sinusoidal patterns” are seen across species at their segment joints and use this as a basis for the generalizability of their model. At present, this claim appears exaggerated because, if I have understood correctly, the authors have analyzed only one gait cycle for one individual from species other than *Polypterus senegalus* (and only one gait cycle for each of four individuals for *P. senegalus*). These small sample sizes make it difficult to ascertain whether the data represented are indeed indicative of consistent patterns, especially for the species other than *P. senegalus*. The range of speeds in the four *P. senegalus* individuals makes their case reasonably convincing, but an increased “sample size” all around (even if this is simply more gait cycles from each individual) will help support the assertion above. Further, it would allow presentation of an average pattern for each species, which likely better represents general patterns (e.g. are the two peaks in the lungfish trace for θ_2 commonly observed? Or a quirk of this particular gait cycle [or individual]? Or perhaps a consequence of their use of the head as a prop rather than their pectoral fins?).

We thank the reviewer for this question and have added additional biological data to alleviate some of the concerns around small sample size. We now have included data from six bichir specimens, four snakehead videos, five catfish videos, two lungfish videos, and unfortunately still only one sculpin video.

These new datasets have been incorporated in a new version of Fig. 4, in which each of the cycles tracked is plotted in a faded color and the mean of each cycle is plotted as a bolded line. Additional plots separating out the results for each category of fish are included in the supplementary material. Finally, the relevant Methods section has been updated.

Fig. 4: Comparison of the walking kinematics of five fish species exhibiting the undulating tripod gait. a) Illustrations of the five species - the African lungfish *Protopterus annectens*, the grey bichir *P. senegalus*, several species of catfish (*Pterygoplichthys disjunctivus*, *Pterygoplichthys multiradiatus*, *Pterygoplichthys gibbiceps*, *Hoplosternum punctatus*, and *Hoplosternum* sp.), the northern snakehead *Channa argus*, and the tidepool sculpin *Oligocottus maculosus*. b) Angular position of the first body joint over the course of one gait cycle for each of the fish species. c) Angular position of the second body joint over the course of one gait cycle for each of the fish species. The lighter traces denote each gait cycle tracked and the dark bolded traces are the average of the tracked gait cycles. The number of specimens for each are $n=2$ (lungfish), $n=6$ (bichir), $n=5$ (catfish), $n=4$ (snakehead), $n=1$ (sculpin). d) Diagram of the discretization overlaid on an illustration of the bichir where the dotted line denotes an extension of the middle segment to define the angles θ_2 and θ_3 . e) Gait diagram of the walking locomotion of the five species divided into the movement of the tail and the propping anterior contact between the fish and the surface. More details can be found in the Supplementary Information.

We edited the text to describe the collection of additional gait cycles (page 7, paragraph 2):

We tracked the midlines (see Methods Section 4.1) of five different fish that exhibit the undulating tripod gait: *P. senegalus*, the African lungfish *Protopterus annectens* [33], several species of catfish *Pterygoplichthys disjunctivus*, *Pterygoplichthys multiradiatus*, *Pterygoplichthys gibbiceps*, *Hoplosternum punctatus*, and *Hoplosternum sp.* [34, 39], the northern snakehead *Channa argus* [35], and the tidepool sculpin *Oligocottus maculosus* [36] (Fig. 4a). We split the continuous walking behavior into individual gait cycles starting and ending with the tail to the right side of the fish's body and averaged the per-cycle observations.

We added additional details to Section 4.1 of the Methods (page 20, paragraph 1):

For videos containing *Polypterus senegalus*, the positions of the snout, tail, and proximal and distal fin landmarks were tracked using DeepLabCut [58]. DeepLabCut is a markerless pose-estimation framework that employs a convolutional neural network to track user-defined anatomical landmarks. We trained a ResNet-50-based network [59] using twenty manually labeled frames per video, selected automatically using a combination of k-means clustering and uniform sampling. Data augmentation was performed using the “imgaug” framework [60]. The network was trained for 1,000,000 iterations using all videos included in the dataset. Tracked points were visually inspected, and frames with low tracking confidence were excluded from downstream analyses.

And also here (page 20, paragraph 2):

To assess the similarity of our model to the kinematics of four additional types of walking fish (sculpin, snakehead, lungfish, and catfish), we digitized their movements using publicly available video recordings [33-36, 39]. Due to variability in video quality across publicly available recordings, automated tracking was not feasible for all datasets. In these cases, midline coordinates were manually digitized using CurveMapper, a custom MATLAB-based tool for extracting and fitting body midlines from video data (MATLAB R2024b; MathWorks, Natick, MA, USA). Digitized midline points were interpolated using a spline to generate 200 evenly spaced two-dimensional body coordinates per frame, parameterized from the tip of the snout (0) to the tip of the tail (200). Each video was rotated such that the fish moved consistently from right to left within the frame. We manually identified the starting and ending frames of individual gait cycles, where the tail of each fish was pointing upward in the frame (i.e., pointing to the right of the fish's body). We then applied the same processing pipeline used for *P. senegalus*, including downsampling, when necessary, to obtain between 10-13 midlines per cycle, and manual labeling of the midline and relevant anatomical landmarks. We did not consider cycles where the fish was to the extremities of the frame or when it did not walk straight, as turning behavior likely biases the kinematics. To average the results of all the gait cycles for each fish, we linearly interpolated

between the measured data points such that all gait cycles had one hundred equally spaced points between the start and the end of the gait cycle. There were two cycles from two videos of lungfish, nine cycles from six specimens of bichirs, twenty-two cycles from five videos of catfish, fourteen cycles from four videos of snakeheads, and one cycle from one video of a sculpin.

We split the data based on the types of fish in Supplementary Figures S1-5 to more clearly show the breadth of kinematics observed:

Figure S1: Tracking body joint positions individual gait cycles of catfish.

Figure S2: Tracking body joint positions individual gait cycles of bichirs.

Figure S3: Tracking body joint positions individual gait cycles of lungfish.

Figure S4: Tracking body joint positions individual gait cycles of snakeheads.

Figure S5: Tracking body joint positions individual gait cycles of a sculpin.

We also added the new tracked data from *P. senegalus* to Fig. 3e-f that describes the fitting of the base sinusoid parameters for the robotic model. We have also changed the

start of the cycle (tail pointed to the right of the body) to put these results in phase alignment with the rest of the data.

Finally, we have also added the additional data points from the additional *P. senegalus* experiments to Figs. 5a and 6. Fig. 5a is reproduced below and Fig. 6 is reproduced further down in response to one of the specific comments.

Fig. 5a

To help improve the biological dataset the authors could: make use of additional gait cycles already captured (publicly available videos used to collect the current data often have multiple gait cycles), collect new recordings (if animals are available for experimentation), or procure videos from additional (publicly available) previous accounts of the terrestrial locomotion for these species (*Polypterus senegalus*: Standen et al., 2014; Standen et al., 2016; Foster et al., 2018 | *Protopterus annectens*: Horner and Jayne, 2014 | *Hoplosternum* sp.: Bressman et al., 2021).

We thank the reviewer for these suggestions and have included analysis of the supplementary videos provided in Bressman et al (Fig. 4). We have also added additional data from new experiments we performed on *P. senegalus* specimens. The supplementary

video in Horner and Jayne, 2014 is of one cycle beginning with the tail facing to the left instead of the right as we have defined it so we unfortunately could not incorporate it into our analysis.

The mechanism of movement also varies between these species. There are at least two notable differences that should be mentioned in the paper as they likely influence the nature of contacts between the animal and the substrate: in *Protopterus annectens* the “prop” is the anterior of the skull, and in *Hoplosternum* the pelvic fins (and mouth) also contact the ground and potentially constrain the contact points on the animal. These differences in locomotor structures likely impact the kinematics and contribute to differences in Figure 2 (and perhaps change the “optimal”, or perhaps appropriate, segment lengths for these different species).

We thank the reviewer for this comment and agree that these are important points to include as the exact instantiation of the model will indeed differ depending on an individual species’ morphology and propping points.

The propping point does not affect the tracking of the axial column bending that we performed (Fig. 3e-f and Fig. 4b-c). For *P. senegalus*, it appears that both 1) propping occurs via contact between the pectoral fins and the surface and 2) no bending occurs further anterior than the pectoral girdle at which we have labeled the head joint. While this is true for *P. senegalus*, there are certainly other species for which joint and the propping point will be separate. To fully characterize the appropriate segment lengths for each of these species, we would need to run a fitting algorithm on videos of each of the species; using the 2:4:5 segment ratio identified for *P. senegalus* is a good first approximation for the other species but can likely be improved. We leave an investigation of the optimal segment lengths for each of these species to future work, however. We have presented simulations examining the sensitivity of the model to different segment lengths (Fig. 8a) that indicate the model still generates forward locomotion with different ratios of segments. We also examined one case of varying the propping point without varying the segment lengths (Fig. 8a), though this case uses some implausible morphologies.

We have added a line to the discussion of the limitations of the model as needing to tune each instantiation of the model to represent a different species is not ideal but is necessary (page 18, paragraph 5):

Though the undulating tripod gait model is generalizable to many species, both the propping points and the segment lengths need to be adjusted to represent the mechanics of each individual species.

Specifically regarding the propping points and contact constraining the kinematics - we have added a line to Section 2.2 stating that the propping point varies between species (page 7, paragraph 2):

We took the anterior point of the head as the primary propping point for *P. annectens* and the pectoral fins as the primary propping points for the other species.

And later in the same paragraph:

Although the timing of individual movements and the location of the propping structure varies among species and specimens, the overall sequence of prop, bend, and pivot remains consistent within the undulating tripod gait.

We also added text to Section 2.3 describing the propping appendage in the simulation (page 9, paragraph 2):

Since this simulation is inspired by *P. senegalus*, we represent the propping appendages as a rigid bar at the base of the head segment, approximating the pectoral fin contacts; to model other species using different props, this structure would need to be adjusted.

Although we do not address it rigorously in this work, changing the attachment point will change the moment generated by the posterior contacts and reaction forces as the distance and orientation of the forces with respect to the contact points will change. This is tangentially addressed in Fig. 8a as two of the improbable morphologies have the same body segment proportions but different propping locations - we see that in these cases, when contact is made further anterior, it significantly decreases the speed (see the latter two cases labeled "extreme morphology").

Finally, we added analysis of the posterior contacts between the fish and the surface to the potential future extensions of this work (page 19, paragraph 2):

In particular, further analysis of posterior body–substrate contact is needed, including quantifying the magnitude and direction of locomotory forces generated by the tail and determining whether pelvic fin contact constrains motion or alters ground reaction forces.

Further, the substrate across which the animals are moving will impact the physical forces in the system and can affect the kind of movement the animal is capable of (e.g. Standen et al., 2016 and Bressman et al., 2019). This is beyond the scope of the paper to address in a modelling or robotic context, but it bears mentioning with regards to the biological datasets as the type of surface and physical properties of the surfaces that each species is moving across is different. For example, if the authors used movement of *Polypterus* on gravel as their exemplar (Standen et al., 2016), they would likely have reached a dramatically different conclusion.

We thank the reviewer for pointing this out and we agree that the surface affects the gait that each species exhibits. We also agree it's unclear whether the locomotion of the same species on different substrates would still fit the undulating tripod gait paradigm as the fish may need to locomote differently to gain sufficient traction or it may be beneficial to alter the gait to take advantage of other surface properties. We have added some discussion on this (page 18, paragraph 3):

The general undulating tripod gait can be made optimal to specific cases by selecting parameters based on details of each fish species (e.g., specific anatomical features) and its interaction with the environment (e.g., the friction between the skin and the substrate [54] or the terrain it walks on [14, 15]).

We also have listed the need to explore the effect of realistic surface properties on the performance of the undulating tripod gait in the Discussion (page 19, paragraph 2):

It would also be valuable to integrate models of energy cost, substrate interaction, and environmental feedback to assess gait efficiency and robustness under ecologically relevant conditions, while explicitly accounting for real-world surface properties such as compliance, viscoelasticity, and fluid layering.

Finally, some of the videos used for data collection suffer either from the angle of the camera (not directly above the fish) or from the type of lens (fish-eye lens), yet there is no mention of these challenges in the methods. Both of these video characteristics have the potential to influence and/or bias the kinematic (especially midline) data collected. Was any correction used for these characteristics? If so, what corrections were used? If not, the authors need to at least be transparent so that the reader can assess for oneself the level of confidence in the videos. That being said, there is at least one cycle of

movement from a view directly above or directly below a fish in publicly available (even without contacting authors) videos between the references already used and those I have listed above. I urge the authors to use videos that will yield the most reliable kinematic information available, or to be transparent about the ways that the data may have bias.

We thank the reviewer for this comment. In the updated datasets we have collected, we have endeavored to balance the need for larger sample sizes with the quality of videos used and have discarded or cropped the length of some videos in which the fish is turning, or away from the center. We used the same procedure for all the data collection to be consistent as in our estimation, the additional compensation for potential lens effects likely would add other unknown noise to the measurements.

As suggested, we have added a Supplementary Information section S1.1 briefly mentioning these factors in the videos used:

In this work, we rely on videos of walking fish available online and from previous work. We attempted to balance the need for a significant sample size of gait cycles with the suitability of the videos for analysis. As such, we discarded videos or cropped the length of videos where the fish is walking in a curved path, close to the extremities of the frame, or when an extreme fish-eye distortion was present. However, we note that one video of *Hoplosternum* sp. [34] is noticeably not from an overhead view (angled view), one video of *P. annectens* [33] has some noticeable shake, and two videos of *C. argus* [35] were taken using a fish-eye lens, though we cropped these videos to get gait cycles close to the center of the field of view.

MINOR COMMENTS

Lines 42-50: The authors have outlined a mechanism of generating movement that relies on “coordinated axial and appendicular movements”, however the first example they provide (mudskippers) are generally understood to use very little (if any) body movement to generate forward locomotion. It seems like the focus of this section of text is the use of ground reaction forces rather than hydrodynamic forces during walking and the diversity of walking strategies, so I recommend re-wording the first sentence (“On land, walking fish generate...”) to include all possibilities listed in the subsequent examples (e.g. “On land, walking fishes generate forward motion through axial movements, appendicular movements, or the combination thereof...”). This would avoid any incorrect implications in the examples described.

We thank the reviewer for the comment and agree that this should be changed. We have edited the sentence (page 2, paragraph 1) as follows:

On land, walking fishes generate forward motion through axial undulation (motion of the vertebral column), appendicular articulation, or a coordinated combination of both, relying on ground reaction forces rather than hydrodynamic thrust [7].

Lines 54-62: Here I find it somewhat confusing that the authors assert that “...fishes similarly blend axial bending with appendage-based support in ways that defy simple classification...” but then provide examples of locomotion that fit rather neatly into those classifications without any further explanation of their proposed “continuum”. Current consensus (e.g. reviewed in Pace and Gibb 2014 and Lutek et al. 2022) in the literature is that there are indeed species that use axial movements (e.g. ropecod and eels; perhaps primarily, though it is unclear if the fins provide additional propulsion in these cases), appendicular movements (e.g. mudskippers; perhaps primarily, but contributions of the body are unclear), and many that use a combination of axial and appendicular movements (the focus of this paper). I agree that there is likely extensive variation both within and between species for the axial-appendicular “mode” of locomotion in terms of the “contribution” of appendages and body (and perhaps there is more variation than has been acknowledged in the literature for the axial and appendicular “modes” as well), but whether this is indeed a “blending” that leads to continuity is still an open question, rather than a demonstrated characteristic of terrestrial locomotion in fish (and indeed, the authors really provide no evidence to support their claim of “continuity” in a terrestrial environment – the examples they use fit quite well into three distinct categories). I would suggest that this framing of terrestrial locomotion either needs some clarification, including references to support the assertion of “continuity” in the terrestrial environment, or I would suggest that the authors instead focus on the unifying “physical principles” in a terrestrial environment – these principles are indeed, as the authors point out, understudied and apply across the listed “modes” (and in both aquatic and terrestrial environments) regardless of whether the locomotion is a continuum or discrete categories. For example, the following version of this excerpt would shift the focus away from the continuum assertion (which really isn’t the focus of this paper anyway): “Rather than falling into discrete kinematic categories, swimming behaviors form a continuum shaped by the interplay of body form, flexibility, and flow characteristics [10–12]. Similar interplay is likely involved on land as walking fishes can use both axial bending and appendage-based support which may be governed by shared, substrate-independent mechanical principles. Some species have been described to primarily articulate their pectoral fins (e.g., mudskippers [13]), some to primarily articulate their vertebral columns (e.g., ropecod [14]), and some to use a combination of both (e.g., bichirs [15]). However, these categorizations use kinematic observations to classify the gaits of different species but do not explore the underlying physics principles of the gait.”

We thank the reviewer for this comment and appreciate this framing. The reviewer is correct that mudskippers and ropecod belong on the extreme edges of the gait space and that there is extensive variation in gaits exhibited by the species that use both axial and appendicular

articulation. Here, we want to express the idea that describing this large group of remaining species as “axial-appendicular” is not a particularly helpful classification because of this extensive variation and thus far there are no convenient simple groups to further divide “axial-appendicular”.

We have rewritten this paragraph (page 2, paragraph 2) to refer to a “combination” of axial and appendicular usage rather than “continuum” or “blending” as these terms might imply something that we do not intend:

With regards to terrestrial locomotion, previous work has instead attempted to classify walking fish using discrete categories such as using axial-bending, appendage-based support, or an axial-appendicular combination [7]. Of these, a few species employ almost entirely appendage articulation (e.g., mudskippers [13]) or almost entirely axial motion (e.g., ropefish [14]), but the vast majority of fishes exhibit both to a significant degree. It is difficult to further divide this main group into meaningful categories because of the extensive variation in locomotory kinematics between species. Species also exhibit different usages of their vertebral columns and pectoral fins depending on the environment in which they are moving [14, 15], introducing further combination of these characteristic motions. Classifying the gaits of different species in this manner using these axial-appendicular categorizations relies on kinematic observations without exploring the underlying physics principles of the gait.

Lines 90-94: The figure legend should list the species for which the snapshots are provided, where these snapshots are from, and how the snapshots were selected.

We thank the reviewer for this suggestion. The species are the same as those included for analysis later in the paper (e.g., Fig. 4). The individual snapshots that were drawn were representative snapshots of each species as these individuals took visually similar postures. As such, this figure is meant to be illustrative and generally informative rather than to be taken as quantitative data (this is also why there are no scale bars here).

The sketches shown are of representative body shapes taken throughout one gait cycle to illustrate the visual similarity of the kinematics of the following fish species: the African lungfish *Protopterus annectens* [33], the grey bichir *Polypterus senegalus*, the armored catfish *Hoplosternum sp.* [34], the northern snakehead *Channa argus* [35], and the tidepool sculpin *Oligocottus maculosus* [36].

Lines 107-112: The figure legend should include some description of where these snapshots are taken from (I assume the top panel is spaced evenly throughout a locomotor cycle, but I’m not sure when the bottom panel snapshots represent).

We thank the reviewer for this comment. Because of the variability between the species that exhibit the undulating tripod gait, the images are not of any specific time step. For example, a species with a higher amplitude of oscillation will reach the intermediate

steps at different fractions of the gait cycle than a species with a lower amplitude of oscillation. The only guarantee is that the start and ending image is at the start and end of the gait cycle.

To make this more clear, we have added a line to the text (page 3, paragraph 4):

The timeseries of postures within a gait cycle depends on the exact morphology and gait parameters; rather, it is the sequence of these general actions that defines the undulating tripod gait (Fig. 1).

And added a line to the Fig. 2 caption.

The schematics are not linked to specific timesteps as the postures of the fish within its gait cycle depend on the exact gait parameters.

Lines 115-122: Perhaps a reference to Figure 1 would be helpful here since you have nicely illustrated this behaviour here.

We thank the reviewer for this suggestion and have added it at the end of the paragraph (page 3, paragraph 4).

Lines 178-179: Because of the large variation in the range of theta in Figure 4b and 4c it might help the reader focus on patterns of angle fluctuation (“sinusoidal patterns”) to normalize the traces for each species to their maximum and minimum (such that values range from -1 to 1). Especially since the authors are focussing on phase relationships, this would help readers visualize the relative timing of maximum and minimum angles. If the biological dataset is augmented (as I suggest above) then I would also recommend directly presenting the phase relationship between θ_2 and θ_3 . While I agree in principle that a conserved phase relationship is evidence that all of these walking gaits could be described by a single model, the data as currently presented does not sufficiently support this assertion.

We thank the reviewer for this comment and agree that the phase relationship is less clear than the alignment of the overall timeseries of θ_2 and θ_3 trajectories. The variation in phase is most clear in the motion of the catfish specimens, which also exhibit the smallest magnitude of body bending.

Part of the goal of Fig. 4b and 4c is to indicate that one of the principal differences between the gaits of different fish is the variation in the body bending, and we think that normalization would remove that representation of the amplitude of θ_2 and θ_3 . Instead, to more closely show the phase relationship, we have included Supplementary Figures S1-S5 plotting of all of the tracked gait cycles for the five types of walking fishes (see response to Major Comment 1) and we have fit the sinusoidal parameters specifically to the kinematics of *P. senegalus* in Fig. 3e-f and Table 1.

Lines 184-193: At present it is unclear the sample sizes for the line plotted in Figure 4b,c and e. Please specify whether the data is an average or from a single gait cycle.

We thank the reviewer for this comment and have added this to the Figure 4 caption:

The lighter traces denote each gait cycle tracked and the dark bolded traces are the average of the tracked gait cycles. The number of specimens for each are n=2 (lungfish), n=6 (bichir), n=5 (catfish), n=4 (snakehead), n=1 (sculpin).

Lines 192-193: Please clarify whether the “anterior contact point” in the gait diagram (Figure 4e) is the “prop” or the anterior body contact.

We thank the reviewer for this comment as it appears our terminology is confusing. In our model of the undulating tripod gait, the anterior body contact is always used as the prop, so we have used the terms “anterior body contact” and “prop” interchangeably. We have added the word “propping” here (Figure 4 caption) to resolve this:

e) Gait diagram of the walking locomotion of the five species divided into the movement of the tail and the propping anterior contact between the fish and the surface.

Lines 247-249: Can the authors provide either video evidence or quantification of this such that readers can assess this claim?

We thank the reviewer for this comment and we have edited the text for clarity and added the correct figure reference to the text (page 10, paragraph 2):

The measured speed of *P. senegalus* was slower than that of the simulated robot, although the discrepancy between the speed of the simulated robot and the real fish was smaller at higher frequencies than at lower frequencies (Fig. 5a).

Lines 264-267: I wonder if the *P. senegalus* data points would be more useful if plotted at the specific amplitude for that gait cycle, rather than at the average θ_2 and θ_3 amplitude. Presumably a sine curve could be fit to each analyzed gait cycle (or the amplitude could be determined empirically) and that this would provide a more precise comparison between these values for the fish and those for the simulation.

We thank the reviewer for this comment. With the additional *P. senegalus* experiments described in Major Comment 1, there is a large variation of values at lower frequencies and walking speeds, but there is nice matching at the higher speeds. We have edited both Fig. 6a and 6b to include this new data:

We have also added the following to the Methods section to briefly describe the sinusoidal fitting (page 21, paragraph 3):

We used a least squares method to fit the tracked joint angles in the *P. senegalus* experiments to sinusoids of the form $\theta = A \sin(2\pi x + \phi) + C$ where A is the amplitude, x is the normalized fraction of the gait cycle, ϕ is the phase, and C is a constant offset. Here, we fixed the frequency to 2π as we have defined each set of data as a complete cycle.

Lines 304-306: The panels (a) and (b) in this figure are not correctly identified in the figure caption. Additionally, as for Figure 6, I believe it would be more appropriate to plot the *Polypterus* values at the measured phase for each gait cycle.

We thank the reviewer for this comment as the subfigure captions are indeed swapped. We have corrected the Figure 7 caption as follows:

a) Gait diagram of *P. senegalus* for observations of forward and backward walking. b) Speed and direction of the simulated robot for different phase offsets

Regarding plotting actual phases measured from *P. senegalus*: unlike just tracking the midline to discretize for the two angles angles that is mostly clear from the dorsal view, we find that there was too much noise between the tracking of the fins in the dorsal view (difficult to assess exact contact time from that view) added to the noise of tracking the midline and discretizing the body into segments.

We have added Supplementary Table S4 that details the average per-cycle sinusoidal fittings from the *P. senegalus* experiments. The mean difference in phase between the head and tail joints is 1.4 with a standard deviation of 0.31.

Lines 307-326: Including additional figure references throughout this section where appropriate (and perhaps including a figure that depicts the relationship between speed and ϕ_3) will help clarify the statements here.

We thank the reviewer for this comment and have added figure references to the paragraphs listed:

Page 12, paragraph 3:

During observations of *P. senegalus* specimens, we noticed that the coordination between the pectoral fin contact and the tail motion differed between forward and backward walking (Fig. 7a and Supplemental Movie S4). This was also reflected in the simulations varying phase where the robot consistently moved forward when $\phi_2 = 0$ or $\phi_2 = \pi/2$ and consistently walked backward when $\phi_2 = \pi$ or $\phi_2 = 3\pi/2$. In addition, the simulated robot moved forward fastest when $\phi_2 = \pi/2$ and $\phi_3 = \pi$, which were the same parameters observed in the fish's locomotion (Fig. 7b).

Page 13, paragraph 3:

However, the maximum forward and backward walking speeds were achieved with $\phi_3 = \pi$, while $\phi_3 = 0$ led to the worst locomotion (Fig. 7b).

Lines 335-350: Include figure references throughout these paragraphs to direct the reader to where this data can be found.

We thank the reviewer for this comment and have added figure references in Section 2.4.4:

(page 13, paragraph 5):

0.27 BL/s (Fig. 8a)

(page 14, paragraph 1):

effectively combining the middle and tail segments into a single rigid body (Supplementary Movie S7).

(page 14, paragraph 1):

6.1% of the maximum forward speed with three body segments (Supplementary Table S3).

We have also added Fig. 7c to present the data in Fig. 7b organized as a function of ϕ_3

b) Speed and direction of the simulated robot for different phase offsets as a function of ϕ_2 . c) Speed and direction of the simulated robot for different phase offsets as a function of ϕ_3 .

Lines 411-415: Unclear if a “full factorial” set of combinations (i.e. all possible combinations) was used for these tests, or if a subset were tested. It is therefore also unclear what combination of surface and fin frictions is presented in Figure 9f.

We thank the reviewer for the comment. All possible listed combinations were tested; however, one case (high fin friction and high surface friction) resulted in no forward motion. In this case, the high friction caused the robot to roll; there must be some net slipping between the robot and the surface otherwise the only solution to the kinematic constraints is rolling.

We have added a statement to the text (page 16, paragraph 1) to indicate that all combinations were tested and one of the four combinations could not create stable locomotion:

While we tested the robot with both high and low friction fins on both high and low friction surfaces, the robot was not able to move with high friction fins on the high friction surface as it consistently rolled over

We also added some context to the Fig. 9 caption:

f) Comparison of the speed of the robot on different surfaces. "Base" denotes the robot with high friction fins on the low friction surface ($\mu_s = 0.30$), but the robot was not able to walk with the high friction fins on the high friction surface ($\mu_s = 0.38$). "Low fr fin" denotes the robot with low friction fins on the low friction surface ($\mu_s = 0.27$), and "high fr surface" denotes the robot with low friction fins on the high friction surface ($\mu_s = 0.32$).

Line 433: I would caution against the use of the word "efficiency" here – the authors have not actually measured any values that quantify efficiency (e.g. cost of transport) but rather have quantified the speed of movement. Perhaps an argument could be made for "peak locomotor performance", although "performance" is also a loaded term that usually requires measures beyond just speed. Perhaps the most precise and appropriate phrasing would be "peak locomotor speed".

We thank the reviewer for the suggestion and agree that this work primarily uses locomotor speed as the output of interest. We have made the change as suggested (page 17, paragraph 1):

Our results show that the gait observed in *Polypterus senegalus* occupies a region of the parameter space associated with peak locomotor speed,

Line 436: Perhaps I've misunderstood, but I believe the results presented also suggest that frequency also modulates locomotor speed (Figure 5).

We thank the reviewer for this comment. The reviewer is indeed correct, frequency also modulates locomotor speed and we have added that to this sentence (page 17, paragraph 1):

while the walking speed is modulated by both the amplitude of axial bending and the frequency of undulation

Line 439: I believe this should read "... morphology of *P. senegalus*, aligned closely with..."

We thank the reviewer for the attention to detail and have corrected the wording of this sentence (page 17, paragraph 1):

Finally, we found that the fastest walking gait of both the simulated and physical robots derived from the morphology of *P. senegalus* aligned closely with the empirical kinematics observed in live animals.

Lines 449-451: While not perhaps exactly the same as the walking discussed here, the idea that "many fishes may possess latent terrestrial capabilities" has been investigated for jumping in several species of fish and would perhaps be worth a mention and/or

citation here. See as a starting point Gibb et al., 2011 and Gibb et al., 2013. This would also help make the discussion more rich and well grounded in the literature.

We thank the reviewer for suggesting this connection and have added some text to reflect this idea (page 17, paragraph 2):

The ability to exploit these interactions without substantial morphological innovation may reflect a broader functional plasticity in fishes via which latent terrestrial capabilities may be expressed under specific environmental conditions (e.g., hypoxic stress, predator evasion, seasonal habitat shifts, or movement between tidal pools) where temporary access to land or structured substrates can confer survival advantages [40, 42]. Similar hypotheses have been posited regarding fish that exhibit jumping behaviors without specialized anatomy [43], providing additional evidence that fish can generate different kinematics as needed on land.

Line 458: It is unclear to what the authors are comparing the magnitude of muscular force here. For the few species that have been studied, terrestrial locomotion tends to require higher magnitude muscle activation (which presumably produces higher forces) than those used for aquatic locomotion (e.g. Eels, Polypterus, Rivulus, Lungfish; reviewed in Lutek et al., 2022). If the authors are comparing this theoretical muscle output to what would otherwise be required for terrestrial locomotion, this should be made explicit.

We thank the reviewer for pointing out that this is unclear. Here, we are comparing the terrestrial undulating tripod gait to other terrestrial stepping gaits like those of the mudskipper, not to aquatic locomotion. Stepping gaits require the pectoral fins to generate work, whereas the undulating tripod gait does work with the tail and only uses the anterior contact as a prop. While we do not rigorously investigate this, it is plausible that the musculature of the tail generally used for swimming is more easily adapted to generating force for terrestrial locomotion than the pectoral fins are.

To clarify this, we have changed this section of text (page 17, paragraph 3):

By relying on passive ground reaction forces, stable axial resonance, and low-complexity joint trajectories, the undulating tripod gait minimizes the need for precise motor control of stepping or high force output from anterior structures (e.g., pectoral fins) compared with more specialized gaits [8, 9].

Lines 458-459: "...specialized stepping gaits that involve active body lifting..." See, however, Standen et al., 2014 - Polypterus raised on land do indeed lift their bodies (as measured by nose elevation) higher off the ground than fish raised in water, suggesting that at least for the focal species of the present manuscript lifting of the body off the ground is indeed an important part of this kind of "gait" on some substrates. It would

also help strengthen this statement to provide examples (and appropriate references) for the “specialized” gaits to which the authors are referring.

We thank the reviewer for flagging this unclear point. The paper by Standen et al does indeed show that fish that are raised in terrestrial environments lift their bodies, but that fish raised in aquatic environments do not, so this example does involve a bit of an anatomical specialization. In this work, our analysis is of aquatically raised *Polypterus* specimens and our model of the undulating tripod gait does not include a specific joint for elevation of the head.

However, the point we are trying to make here is more about work done by the pectoral fins. In the undulating tripod gait, the pectoral fins are used just as props for the pivoting motion whereas in other (stepping) gaits, they are used for work output. To attempt to reduce confusion about the main point here, we have removed the phrase “active body lifting” and instead describe it as “high force output from anterior structures” (page 17, paragraph 3):

By relying on passive ground reaction forces, stable axial resonance, and low-complexity joint trajectories, the undulating tripod gait minimizes the need for precise motor control of stepping or high force output from anterior structures (e.g., pectoral fins) compared with more specialized gaits [8, 9].

Lines 460-462: The statement “This may be particularly advantageous in...” is absolutely true in theory, but at present there is little evidence (or perhaps “conflicting” evidence) for whether walking (or for that matter terrestrial locomotion as a whole) in fish is anaerobic. Many amphibious fish species possess specializations that enable respiration on land. In fact all of the example species used in this paper have specializations: *Polypterus* have a lung, lungfish have a lung, armoured catfish have an accessory respiratory organ, snakeheads have a suprabranchial organ, and tidepool sculpins can perform cutaneous respiration, and there are various accounts that support both the use of anaerobic (e.g. eels white muscle activity tends to increase on land; Gillis 2000, *Polypterus* pectoral fin fiber type switches to “fast” fibers; Du & Standen, 2017, Mangrove rivulus have higher “body muscle” activity on land; Perlman and Ashley-Ross, 2016) and aerobic (e.g. red body muscle activity tends to be higher on land than in water for *Polypterus*; Foster et al., 2018, Mangrove rivulus increase the size of red, but not white muscle fibers after terrestrial acclimation; McFarlane et al., 2019) muscles for terrestrial locomotion. This makes it difficult to assert with any certainty that the systems investigated are using anaerobic mechanisms to move. This is indeed an interesting possibility and would benefit from a more complete discussion of this idea and the implications of the author’s model when interpreted in the context of the whole of the terrestrial fish locomotion literature.

We thank the reviewer for this comment and appreciate the sources provided. The idea is more about efficiency than about the exact mechanism (aerobic or anaerobic) that is being

used. Our point is that it benefits the fish to minimize the energy usage on land if possible. If the undulating tripod gait enables locomotion without additional specific body elevation against gravity or without needing to produce large amounts of forces with relatively weak appendages, it could have an energetic advantage. While we believe that energy efficiency could be a potential benefit of the undulating tripod gait, we have not rigorously explored that in this work.

We have rephrased this section as follows to include this as just one example in a very broad discussion of how the undulating tripod gait may benefit the energetics of walking fish (page 17, paragraph 3):

Moreover, the simplicity of the gait may offer energetic advantages. By relying on passive ground reaction forces, stable axial resonance, and low-complexity joint trajectories, the undulating tripod gait minimizes the need for precise motor control of stepping or high force output from anterior structures (e.g., pectoral fins) compared with more specialized gaits [8, 9]. This may be particularly advantageous in low-oxygen environments or during anaerobic bursts of activity, where reducing postural costs and maintaining mechanical efficiency is essential [44]. Benthic or amphibious species not specialized for terrestrial locomotion could also benefit from the use of the underwater tripod gait as a more stable and energetically efficient means of moving slowly or station holding, as hovering in midwater incurs high energetic costs due to the need for constant fine motor control to counteract mechanical instability [45].

Lines 504-591: Many of these paragraphs are strikingly lacking any grounding in the present literature. While I am more well versed in the literature of the biology of amphibious locomotion than robotics and models, I wonder if there are places where relevant literature would help strengthen the discussion. For example, lines 524-526 discuss the impact of morphology and friction on terrestrial locomotion. I believe this would have been published after this paper was submitted, but Lopez-Chilel and Bressman (2025) have addressed the impact of mucus and scale anisotropy for terrestrial locomotion in fish which seems directly relevant here. Likewise, lines 529-532 would benefit from explicit discussion of (and reference to) the relevant examples mentioned indirectly.

We thank the reviewer for this comment and for the specific suggestion of the paper by Lopez-Chilel and Bressman - we have added this citation to a comment about interaction with surfaces and terrains (page 18, paragraph 3):

The optimal parameters for the undulating tripod gait depends on details specific to each fish species (e.g., morphology) and its interaction with the environment (e.g., the friction between the skin and the substrate [54] or the terrain it walks on [14, 15].

We have also reorganized the discussion section to focus on the main points and to be more concise. Along with that, we have added a number of citations throughout the section so

that we have citations for the statements that are not directly related to the conclusions or implications from the results in this work.

Lines 594-615: Be sure to explicitly mention the sample sizes for each species (number of individuals and number of cycles analyzed).

We thank the reviewer for this comment and have added the following to Section 4.1 of the Methods (page 20, paragraph 3):

There were two cycles from two videos of lungfish, nine cycles from six specimens of bichirs, twenty-two cycles from five videos of catfish, fourteen cycles from four videos of snakeheads, and one cycle from one video of a sculpin.

movieS1 & S4: It would be helpful to have a scale bar to assess speed of the animal, if possible.

We thank the reviewer for the comment and have added the scale bars to Movies S1 and S4.

Reviewer #2 (Remarks on code availability):

I was indeed able to run the code. The README is sufficient, but some further explanation of how to change parameters (or examples of what those parameters should be set at to get the different situations presented in the paper) would be helpful.

We thank the reviewer for the comments on the code repository. We have added an additional readme to the top level folder of the repo detailing what is in each of the setup files, how to adjust the gait and morphology parameters, and how to achieve the gait most similar to *P. senegalus*, which is what the configuration is set up to achieve on download.

The updated repo can be found here:

https://github.com/mishidasan/undulating_tripod_gait_simulink_rev1

Reviewer #3 (Remarks to the Author):

The authors present a study of the locomotion adaptations of walking fishes that engage in the “undulating tripod gait.” They employ simulations, biological observations, and robotic proxy experiments to investigate the impact of gait and morphological parameters on locomotion. The metric of interest, in this case, is forward speed of the animal/robot. The authors systematically vary different parameters, such as the amplitude or phase offset of sinusoidal inputs to the simulated robot’s joints, and measure the resulting forward speed. They compare results calculated, when relevant, with those gleaned from observing the biological organism. Ultimately, the authors state that the simulation/robotic proxies recapitulate biological observations to a reasonable degree. From these, they make remarks about the undulating tripod gait as a simplistic, widely-seen locomotion strategy, and hypothesize about its utility in the transition from aquatic to terrestrial locomotor modes.

The science presented herein is interesting, and the experiments provide a strong foundation for a compelling story. However, a number of issues with the manuscript collectively indicate that it is not yet ready for publication. These issues include: unclear figures, sometimes unclear documentation of methods, insufficiently justified choices for parameter sweeps (prompting many whys? throughout), insufficient amount of biological experimental data (making for very sparse comparisons with experiments and difficult to see trends), insufficient amount of simulation model experimental data, and unconcise/redundant discussions. Among major points are the following:

-Simulation has the ability to sweep massively through the parameter space with virtually no additional expense. Why not look at results for more of these preternatural parameters, pushing to the limits of what would be possible in physical systems? Instead of generating sparse data points with simulation, you have the opportunity to generate “surfaces” that give a much more continuous intuition of how parameters interact in your experiments.

We thank the reviewer for this suggestion and agree that a more continuous exploration of the parameter space could be a very valuable use of this simulation. Our aim for this work is to discuss the undulating tripod gait as a model that can be applied to a number of fish species capable of terrestrial walking. We agree that examining more extreme, supernatural gait parameters and morphologies would be interesting to see the limits of this model. These cases that push the limits of the undulating tripod gait are not biologically realistic, though, and extensive analysis of this moves away from the focus of work on whether our model is relevant specifically to *P. senegalus* and more broadly to other walking fish species.

The data points we have generated span a large portion of the design space in which our biological inspiration *P. senegalus* is observed and it is unlikely that we will find other

significant maxima under the assumptions of this model and the geometries derived from *P. senegalus*. While rigorously exploring the many other species that exhibit the undulating tripod is outside of the scope of this work, we agree that future work should explore the parameter space for other specific morphologies.

We have added some discussion of the link between morphology and performance within the space of the undulating tripod gait (page 18, paragraph 3):

The general undulating tripod gait can be made optimal to specific cases by selecting parameters based on details of each fish species (e.g., specific anatomical features) and its interaction with the environment (e.g., the friction between the skin and the substrate [54] or the terrain it walks on [14, 15]).

We do touch briefly on more extreme morphologies when we simulate robots with implausible body proportions, showing that these morphologies still all generate forward motion with varying degrees of success. The data is plotted in Fig. 8 and we have added some new text (page 15, paragraph 1):

All of the simulated robots with altered morphologies are capable of forward locomotion with the same sinusoidal control parameters applied in previous experiments (Supplemental Movie S5). This indicates that the model is valid for a wide range of morphologies, even including some that are implausible for real fish to exhibit. These example morphologies were chosen to span some of the potential design space of the simulated robot and support the idea that a range of species capable of various degrees of body bending would be expected to exhibit different parameters for the undulating tripod gait (Fig. 4).

-There does not appear to be a lot of cohesion between the parameters evaluated in the simulations, the actual animal, and the robot. More data would seem necessary so plots are not, for example, 10 simulation points and 3 biological points. It would be compelling if (for the cases possible to evaluate on the animal and robot), a large set of parameters was plotted together to show more comprehensively agreements / disagreements between sim/bio/robot.

We thank the reviewer for this comment. We agree that it would be useful to show a larger variety of data points from *P. senegalus* but we are limited to the observations of the specimens we have and have little ability to induce different observations to obtain the range of biological data that is suggested here. To address this, we have included data from additional experiments with *P. senegalus* specimens to generate more biological data. This new data is in line with the previous data of *P. senegalus*, though there is a much wider range of overall walking speeds observed.

We have added relevant additional results to Fig. 5a:

And Fig. 6a and 6b:

-You make interesting observations that the animal seems to fall into some sort of range consistent with the best performing simulations. Is this “locally optimal” parameter range for forward locomotion speed maximization representative? I’m not sure, due to the limited number of experimental datapoints and from the fact that simulation did not cover a broader range in the parameter sweep. Furthermore, are there certain conditions, e.g. fight or flight, wherein the animal would exhibit a faster gait? Do these types of situations, perhaps, when observed in the wild, influence the undulating tripod gait ‘basis,’ deviating it from the nominal case that you studied in the lab?

We thank the reviewer for this comment as these questions could be very useful for guiding future directions of similar research.

First, in this work we have attempted to study the fish at its volitional steady state speed to try to obtain the most consistent results possible, rather than deliberately provoking acceleration or unsteady locomotion. It is plausible that the fish could move faster as a response to danger by increasing the frequency of its undulations (Fig. 5). As such, the limits to this speed are not likely defined by the gait of the fish but by aerobic capacity and muscle strength, neither of which we can measure with the methodology presented here.

Previous work has shown that fish exhibiting what we call the undulating tripod gait will walk differently on different surfaces or when navigating through obstacles, indicating that there are indeed deviations from the gaits we present here to compensate for or take advantage of the different frictional environments. While it is out of the scope of this work, it could be informative to simulate the model in varied environments in which the actual fish might be reluctant to walk, as it is unclear whether the terrain-specific gaits should still be classified as the undulating tripod gait. We have added some discussion on this (page 18, paragraph 3):

The general undulating tripod gait can be made optimal to specific cases by selecting parameters based on details of each fish species (e.g., specific anatomical features) and its interaction with the environment (e.g., the friction between the skin and the substrate [54] or the terrain it walks on [14, 15]).

We also have listed the need to explore the effect of realistic surface properties on the performance of the undulating tripod gait in the Discussion (page 19, paragraph 3):

It would also be valuable to integrate models of energy cost, substrate interaction, and environmental feedback to assess gait efficiency and robustness under ecological conditions, and future investigation into real-world surface properties, such as compliance, viscoelasticity, and fluid layering, is needed.

-A large portion of the discussion section is repeating previous content and long-winded. Better to opt for a more concise description of the takeaways of the present study, rather than a retrospective justification of the methodologies and a rehashing of the take-aways nearly verbatim from prior sections. Everything from "Our findings reinforce the value of minimalistic models and robotic analogs in uncovering unifying principles of locomotion." Especially applies to this comment.

We thank the reviewer for the comment and have rewritten much of Section 3 to address this concern. Because these changes are throughout the entire Discussion section, in the interest of space in this response document, we are not reproducing all the changes here.

The majority of the editing pertaining to concision was done starting with the second paragraph of Section 3 (page 17, paragraph 2).

Consider other points below, which occur in roughly the same order they can be found in the manuscript.

-The term “axial,” as in “axial body undulation” does not give a concrete image, and should be better described from its first mention. “Axial” is also used in the abstract and leaves the reader wondering which “axis” this motion truly lies about. It would benefit from an early illustration, e.g. in figure 1.

We thank the reviewer for pointing out that this could be confusing. Here we are referring to the axis along the body aligned with the skull and vertebral column and “axial” is the standard term in fish biology for this particular concept. To prevent confusion with the coordinate axes in an engineering sense, we have described “axial undulation” in the first instance of the introduction (page 2, paragraph 1) as “motion of the vertebral column”:

On land, walking fishes generate forward motion through axial undulation (motion of the vertebral column), appendicular articulation, or a coordinated combination of both, relying on ground reaction forces rather than hydrodynamic thrust [7].

-“Despite this shared reliance on substrate interaction, species differ markedly in their terrestrial walking strategies.” Seems to contradict abstract statement about remarkably similar gaits.

We thank the reviewer for this comment and we see how this could be confusing for the reader before more context is provided later in the introduction and elsewhere in the manuscript. In the introduction, the species that “differ markedly” are the larger set of walking fish species (some of which primarily use appendicular forces for locomotion and are not similar to the undulating tripod gait), whereas in the abstract, the “remarkably similar gaits” refers to the smaller subset of species that specifically exhibit the undulating tripod gait.

To address this, we have rewritten the first sentence of the abstract to refer to the similar gaits as a subset of all the species of walking fish (page 1, paragraph 1):

A large subset of fish capable of terrestrial walking exhibit strikingly similar gaits despite spanning across the phylogenetic space and having substantial differences in morphology.

-“Despite striking morphological differences, many fishes exhibit convergent patterns of undulatory locomotion. Rather than falling into discrete kinematic categories, swimming

behaviors form a continuum..." Again, seems to contradict/confuse a bit with the first statements.

We thank the reviewer for this comment; similar to the previous comment, here we are referring to the larger space of gaits rather than just the similar instantiations of the undulating tripod gait and have addressed this by changing the first line of the abstract (page 1, paragraph 1):

A large subset of fish capable of terrestrial walking exhibit strikingly similar gaits despite spanning across the phylogenetic space and having substantial differences in morphology.

Furthermore, we have rewritten part of this paragraph to more accurately describe the combination of axial and appendicular usage and to be more clear about how we would like to improve the categorizing of similar gaits and dissimilar gaits (page 2, paragraph 2):

With regards to terrestrial locomotion, previous work has instead attempted to classify walking fish using discrete categories such as using axial-bending, appendage-based support, or an axial-appendicular combination [7]. Of these, a few species employ almost entirely appendage articulation (e.g., mudskippers [13]) or almost entirely axial motion (e.g., ropefish [14]), but the vast majority of fishes exhibit both to a significant degree. It is difficult to further divide this main group into meaningful categories because of the extensive variation in locomotory kinematics between species. Species also exhibit different usages of their vertebral columns and pectoral fins depending on the environment in which they are moving [14, 15], introducing further combination of these characteristic motions. Classifying the gaits of different species in this manner using these axial-appendicular categorizations relies on kinematic observations without exploring the underlying physics principles of the gait.

-The entire bit from "Researchers use animal-inspired models to investigate the principles underlying specific biological phenomena...nature." is fine justification, but is quite generic and long-winded. I'm not sure it helps funnel the scope of the paper, when a succinct justification for the use of robot template models is already given in the previous sentence.

We thank the reviewer for this comment and agree that this paragraph could be pared down. However, we think that the message of the lines that the reviewer has highlighted is important as we need to motivate our approach of making an very abstracted model of a complex organism (more than just complying with *ceteris paribus*), as we will be arguing in this manuscript that a robot of made from four rigid segments is useful for representing many species of fish.

To make this paragraph more concise and to focus on the goal of reduced order models, we have edited it to read as follows (page 2, paragraph 3):

Bioinspired robotics provides a powerful framework for probing the mechanics of walking locomotion in fishes [16] and for generating new biomechanical hypotheses about organismal function [17]. Robots inspired by animals can be used as stand-ins for the biological organisms to make counterfactual predictions about motions that are not directly observed in nature or to obey the *ceteris paribus* principle in which testing the effects of a single feature can only be done while keeping the other features constant [18]. Models inspired by animals can range from anatomically detailed reconstructions that replicate species-specific morphology [19] to simple abstractions that isolate essential features [20]. By abstracting away taxon-specific traits, models can be built around shared mechanical features, enabling the identification of general principles [21] and the development of unifying frameworks across broad phylogenetic and temporal scales [22].

-The term “convergent gait” should be defined for the wide readership of nature communications.

We thank the reviewer for this suggestion and have added a definition (page 3 paragraph 2):

(a gait evolved separately in a number of distantly related species)

-“ The undulating tripod gait is different from other fish gaits that neglect...” How, exactly, is it different? Would help to be more specific here.

We thank the reviewer for this comment and have rewritten the line to further describe the alternative fish gait (page 3 paragraph 2):

The undulating tripod gait is different from other fish gaits that generate locomotory forces primarily from their appendages rather than from axial articulation [8]

-Fig. 2 would be more clear if roll/pitch/yaw axes were formally defined with some coordinate frame.

We thank the reviewer for this comment and have added the common biological definitions (body frame) to align with the Tait-Bryan angles in the caption of Fig. 2. The pectoral fin tilt is in roll along the anterior-posterior axis (i.e., head to tail), while the head and tail joints are in yaw parallel to the dorsal-ventral axis (i.e., back to belly).

The red box indicates the head and pectoral tilt (rotation in roll, aligned in the anterior-posterior direction), the orange circle indicates the head rotation (rotation in yaw, aligned in the dorsal-ventral direction), and the yellow circle indicates the tail rotation (rotation in yaw, aligned in the dorsal-ventral direction).

-“ When the body is propped on the left side of the midline, the fish swings around the pivot point in the counterclockwise direction and vice versa when the body is propped on the right side of the midline.” Are you referring to Fig 2 here? If so, a reference would help.

We thank the reviewer for the attention to detail and have corrected the reference to point to Fig. 2 rather than Fig. 1.

-“In this model (Fig. 2), we represent the propping motion as being generated by a rigid beam rotating around the fish’s head-tail axis (θ_1) such that $\theta_1 > 0$...” For easier interpretation, please label these variables directly on the figure.

We thank the reviewer for this comment and have added the annotations to Fig. 2:

Figure 2. The simplified model of the undulating tripod gait where the soft body of the fish is discretized into three rigid segments that rotate with respect to each other. The blue hexagons indicate the pectoral fins where the dark blue specifically indicates contact with the ground. The red box indicates the head and pectoral tilt θ_1 (rotation in roll, aligned in the anterior-posterior direction), the orange circle indicates the head rotation θ_2 (rotation in yaw, aligned in the dorsal-ventral direction), and the yellow circle indicates the tail rotation θ_3 (rotation in yaw, aligned in the dorsal-ventral direction).

-“Locomotion results from the interplay of gravitational forces, joint trajectories, and substrate interactions.” Self-evident statement; consider removing.

We thank the reviewer for highlighting this statement. The intention was to indicate that these are the only elements considered by this model as there can be additional factors in other types of fish locomotion, such as adhesion. We have edited this sentence to read (page 5, paragraph 1):

In this model, locomotion results only from the interplay of gravitational forces, joint trajectories, and friction-dominated contact with the substrate.

-“In this work, we focus specifically on the terrestrial locomotion of *Polypterus senegalus* as an exemplar of the undulating tripod gait, in which the fish uses its pectoral fins as propping appendages.” Why this particular animal? Why is it a good representative of the gait, out of the others mentioned?

We thank the reviewer for pointing out that we have omitted this context. *P. senegalus* has a large literature base investigating its terrestrial locomotion, including how its behavior differs between terrestrial and aquatic habitats. Furthermore, it has a long, slender body that exhibits a large degree of axial bending, which implies that this species is physically capable of generating a wide range of gait parameters, whereas species with more stiff bodies might be more constrained in the motions they can .

To add this context to the manuscript, we have added two sentences about the suitability of *P. senegalus* for analysis of the undulating tripod gait (page 5, paragraph 2):

P. senegalus is a species of fish capable of surviving in both aquatic and terrestrial environments, and its ability to live on land for extended periods of time has led to a significant literature on its terrestrial walking gaits [27, 37, 38]. Furthermore, its long, slender, flexible body is capable of significant axial bending, which potentially provides a larger feasible space of possible gait patterns compared to fish with stiffer axial columns.

We also show the data across several species in Fig. 4 comparing the locomotion of four other species of walking fish with the locomotion of *P. senegalus* as evidence that this motion is representative of the others. *P. senegalus* exhibits similar coordination between the tail and the pectoral fin contact and similar distribution of body bending as the other species we considered in this work.

The relevant text is in Section 2.2 (page 7, paragraphs 2 and 3):

Across species, all the fish alternate which side of the body makes contact with the substrate and subsequently sweep the tail toward that planted side (i.e., after engaging the left pectoral fin, the tail bends leftward). Although the timing of individual movements and the location of the propping structure varies among species and specimens, the overall sequence of prop, bend, and pivot remains consistent within the undulating tripod gait. The gait pattern is generally symmetric (Fig. 4e).

Similarly, when these species were approximated using the same discretization into three rigid segments (see Methods Section 4.2), the trajectories of θ_2 (Fig. 4b) and θ_3 (Fig. 4c) exhibited consistent sinusoidal patterns. While the amplitude of joint motion varied among species, the phase relationship between the timeseries trajectories of \$\theta_2\$ and \$\theta_3\$ remained conserved. This is strong evidence that the walking gaits of these species can all be described by the undulating tripod gait model, with *P. senegalus* serving as an exemplar.

-“We tracked the midlines of five different species that exhibit the undulating tripod gait: *P. senegalus*, the African lungfish *Protopterus annectens* [34], the armored catfish *Hoplosternum* sp. [35]...” Here, and this also applies to all other “methods,” it would facilitate ease of reading / understanding if a specific corresponding section in the methods was referenced.

We thank the reviewer for this suggestion and have added the following in

Section 2.2 (page 6, paragraph 1):

(see Methods Section 4.1 and Section 4.2)

Section 2.2 (page 7, paragraph 2):

(see Methods Section 4.1)

Section 2.2 (page 7, paragraph 3):

(see Methods Section 4.2)

Section 2.3 (page 9, paragraph 1):

(see Methods Section 4.3)

Section 2.5 (page 15, paragraph 3):

(see Methods Section 4.4)

-Fig. 3 should have a scale bar indicating the size of the organism

We thank the reviewer for pointing out the omission and have added it to Fig. 3b.

The Fig. 3 caption is updated as follows:

b) Timeseries images of one gait cycle with the discretization of the body into three linear segments overlaid. The three-segment discretization is denoted by blue (head segment), green (body segment), and pink (tail segment) lines. Scale bar is 5 cm.

-“ Across species, the fish alternates which side of the body makes contact with the substrate and subsequently sweeps the tail toward that planted side (i.e., after engaging the left pectoral fin, the tail bends leftward).” Point of clarification - if this is saying something to the effect of: “all observed species showed this behavior” (which I think if your intention), it should read “alternate,” since fish is plural. On the other hand, if it’s a single fish type that exhibits this behavior, then you should specify which one.

We thank the reviewer for pointing out this grammatical error. The reviewer’s understanding is correct and we have corrected this as suggested (page 7, paragraph 2):

Across species, all the fish alternate which side of the body makes contact with the substrate

-Unclear terminology: “angular time series”

We thank the reviewer for pointing out unclear language and have changed this phrase as follows (page 7, paragraph 3):

timeseries trajectories of θ_2 and θ_3

-“ representative exemplar.” Redundant wordage

We thank the reviewer for the attention to detail and have removed the word “representative” (page 7, paragraph 3):

serving as an exemplar.

-Fig 4b-c: what's the sampling rate? It seems to result in fairly jagged curves. Can you be sure you're capturing the true behavior if the sampling rate is so low?

We thank the reviewer for this comment. Because we only collected data ourselves for *P. senegalus*, we do not have a consistent sampling rate for all the videos in Fig. 4b and Fig. 4c, but we have at least twelve points for each gait cycle. We have also collected more data from additional videos and gait cycles of the species Fig. 4b and 4c as well as additional experiments for *P. senegalus* which support that we have captured the true behavior.

The new Fig. 4 shows the additional cycles underneath the mean traces:

Fig. 4b and c:

b) Angular position of the first body joint over the course of one gait cycle for each of the fish species. c) Angular position of the second body joint over the course of one gait cycle for each of the fish species. The lighter traces denote each gait cycle tracked and the dark bolded traces are the average of the tracked gait cycles. The number of specimens for each are $n=2$ (lungfish), $n=6$ (bichir), $n=5$ (catfish), $n=4$ (snakehead), $n=1$ (sculpin).

The individual traces are also separated out by type of fish the Supplementary information:

Figure S1: Tracking body joint positions individual gait cycles of catfish.

Figure S2: Tracking body joint positions individual gait cycles of bichirs.

Figure S3: Tracking body joint positions individual gait cycles of lungfish.

Figure S4: Tracking body joint positions individual gait cycles of snakeheads.

Figure S5: Tracking body joint positions individual gait cycles of a sculpin.

Finally, the additional experiments for *P. senegalus* can be found in the response to Major Comment 2.

-“ We used a generic contact model consisting of a spring and damper...” Would like to see more information and explanation about the simulation setup, perhaps in the SI/methods, to help with reproducibility. I am glad to see the code is included in the submission.

We thank the reviewer for this suggestion and have added the following to the Methods section 4.3 (page 22, paragraph 3).

The head of the robot is attached to the world frame with a six degree-of-freedom joint so that the robot can move freely in space and its translation in the world frame is tracked for calculation of the walking speed. The joints are driven by angular position input and torque is automatically calculated by the simulator based on the defined geometry of the body elements and the inertia given by the defined mass. The ground is an infinite flat plane and contact is evaluated between the plane and the rigid fin and body segments based on whether the geometry of the body element intersects with the plane. There is a normal force applied from the plane to the robot at the point of contact defined as a spring force and damping pressing upward opposing the intersection between the body element and the plane. There is also smooth stick-slip friction with static and dynamic elements opposing the motion of the element in contact with the surface. The position and orientation of the robot in the world frame is determined by the robot's interaction with the surface and with gravity based on the posture of the robot, which is defined by the geometry of the rigid elements and the states of the joints. The simulation uses the ode45 simulator with a variable step size (maximum size 0.01 s).

We have also edited the readme files in the repository provided.

-“ We approximated the behavior of the fin joint as a saturating sinusoid, where the joint reaches a threshold position and holds a constant angular position analogous to the fish holding its fin steady against the ground.” Not clear what “holding its fin steady against the ground” means, and how it relates to the locomotion; what is the connection to cyclic motions with no “holding of configurations” involved?

We thank the reviewer for pointing out the lack of clarity here. This phrase describes the portion of the gait that is analogous to the stance phase in legged locomotion, where the pectoral fin is against the surface supporting the fish's weight, in contrast to other phases of the gait where the fin is lifted from the surface and moving. We chose not to use the terms “stance phase” and “swing phase” used to describe traditional bipedal locomotion because this model allows situations where both fins are lifted from the surface but contact is still made between the body and the surface.

To clarify this, we have changed the text (page 9, paragraph 4):

We approximated the behavior of the fin joint as a saturating sinusoid, where the joint reaches a threshold position and holds a constant angular position analogous to the phase of the gait where the fish is not moving its pectoral fin that is in contact with the ground.

-Fig. 5 does not appear to have sublabels.

We thank the reviewer for the attention to detail and have added subfigure labels:

We have also fixed an issue with the labeling in the Fig. 5 caption:

a) Speed of the simulated robot varying frequency (all joints operate at equal frequencies within the trial). b) Distance traveled during one gait cycle for each frequency.

-“ The measured speed of *P. senegalus* was slower than that of the simulated robot, although the discrepancy between the speed of the simulated robot and the real fish was small at higher frequencies (Fig. 5b).” Where is this difference plotted?

We thank the reviewer for this question and apologize for the error in labeling the subfigures. This difference between the observed locomotion speed of the fish and the speed of the robot is shown in Fig. 5a, not Fig. 5b. We have fixed this reference to the figure (page 10, paragraph 2):

The measured speed of *P. senegalus* was slower than that of the simulated robot, although the discrepancy between the speed of the simulated robot and the real fish was smaller at higher frequencies than at lower frequencies (Fig. 5a).

-How are representative geometries for the simulations defined? The radius of the cylinders, for example — these change the inertia of the appendage, as well as the possible contact locations when walking. These, in turn, influence the resulting forward velocity.

We thank the reviewer for this question and agree with the reviewer’s stated effects of geometry on motion. The geometries and physical parameters for the simulation of the

larger robots were determined based on the feasible robot design, including the actual mass of the motors and a realistic placement of the battery. For the simulation at the scale of *P. senegalus* specimens, we used measurements of the real fish to generate the diameters of the body segments and the total length of the body.

To make this clear, we have added a description of the segment geometries to the Methods section (page 21, paragraph 4):

The geometries of the segments in the simulation of the fish-scale robot were based on measurements of *P. senegalus* specimens while the geometries of the components in the larger simulated robot were based on the actual components of the physical robot discussed in Section 4.4.

-For statements like “The simulated robot moves fastest with the amplitudes of $\theta_2 = 1.5$ and $\theta_3 = 2.1$, which are the same gait parameters observed in our experiments with *P. senegalus* specimens.” It would be nice to see a hypothesis, grounded in physical explanations, as to why.

We thank the reviewer for pointing out this lack of clarity. We address the physical explanations about θ_2 and θ_3 in Section 2.4.2. For θ_2 , the explanation is intuitive; based on the kinematics, we find that an amplitude of $\theta_2 = 1.5$ generates the largest step length (approximately 180 degree rotation of the joint). For θ_3 , the explanation is a bit less clear, but we know that as the amplitude increases at constant frequency, the angular velocity of the joint increases (higher power) but for very high amplitudes, the velocity vector of the tail gains more of the horizontal component.

The relevant text is as follows (page 11, paragraph 2):

Because the fins are attached to the head joint, the amplitude of θ_2 affects where the fin is planted with respect to the rest of the body. From the kinematics of the tripod gait, we expect that the most effective gait will be when the limits of the head joint θ_2 oscillation are at $\pi/2$ and $-\pi/2$. If the fin does not slip against the ground, the head will move in an arc around the stationary fin. Then, the maximum possible step size is where the fin is planted directly in front of the robot in the direction of motion. The joint then rotates π radians and plants the other fin directly in front of the robot.

And (page 12, paragraph 1):

With three rigid body segments, the tail segment contacts the ground and creates most of the pushing force, so the oscillation of the tail joint θ_3 affects both the magnitude and direction of the ground reaction force. As the amplitude of the tail joint increases, both the range of motion and the relative velocity of the tail segment increase because of the larger angular sweep at the same frequency

(Supplemental Movie S3). However, at the highest values of tail amplitude, the angle of the tail drives forces that are more horizontal than forward, reducing the effectiveness of the additional tail sweep.

-Statements like “significantly affects the forward speed of the robot” are qualitative in nature, and would be better understood with numbers and some sort of statistical analysis. For instance, what is the ‘sensitivity’ of the parameter vs that of θ_3 -- which one has more bearing, quantitatively, on the performance? Perhaps PCA would be useful here.

We thank the reviewer for this comment. In the experiments with the simulated robot, we have observed the ceteris paribus principle and within each set of experiments, we have only changed one variable (e.g., in Fig. 6a, we have only changed the amplitude of the head joint oscillation). Thus, we are comfortable saying that these modulations of gait parameters do generate significant differences - for example, increasing the amplitude of θ_2 from 0.5 to 1.0 doubles the speed of the robot.

We agree that sensitivity analysis should be considered if altering multiple parameters at the same time, which we have mentioned in the Discussion section (page 19, paragraph 2):

Looking forward, future work should systematically investigate the sensitivity of the model to specific gait and morphological parameters of interest.

-Are quantities like θ_2 truly constant for the real animal throughout its gait cycle?

We thank the reviewer for the question. θ_2 is not constant within a gait cycle of the fish, it oscillates in a quasi-sinusoidal pattern, while the amplitude of the sinusoid can change between repeated gait cycles. The same is true for θ_3 for an amplitude that can be different from that of θ_2 . Precise statistics of these parameters are difficult to ascertain because of noise in the visual tracking, the distribution of the curvature along the fish’s body, nonlinear trajectories that the fish takes, and other inconsistencies inherent in biological behavior.

To address this, we have added a line in the Discussion section (page 18, paragraph 3):

By abstracting away species-specific anatomical complexity, we isolated the fundamental features of morphology necessary for producing forward terrestrial walking, and by approximating the gait as a set of sinusoids, we reduced kinematic noise from biological recordings to enable consistent exploration of the gait parameter space.

-“ When the phase of the head joint is zero ($\phi_2 = 0$), the head segment begins in-line with the middle segment and begins by rotating clockwise with respect to the joint (i.e., the

head rotates to the right of the body). When the..." I think the descriptions of phase should be supplemented by a simple illustration, to help readers grasp their definitions more quickly.

-Fig. 7: are the captions switched? Does not make sense otherwise.

We thank the reviewer for noticing this and have fixed the Fig. 7 caption;

a) Gait diagram of *P. senegalus* for observations of forward and backward walking. b) Speed and direction of the simulated robot for different phase offsets

-“For this set of simulations, we reduced the frequency of the gait to 0.5 Hz because of the larger body size with more inertia.” More information on why would be nice – is this just the physical limit of the hardware you tested on? Are you trying to keep another physical parameter constant?

We thank the reviewer for pointing out the lack of detail here. This is a hardware limitation (maximum angular speed) of the specific motors used that becomes relevant at higher amplitudes of oscillation. We have added the relevant information to the section of the Methods (4.4) discussing the robot hardware (page 22, paragraph 5):

Because the maximum angular speed of these motors was about 1 rev/sec, we reduced the frequency of the physical experiments to 0.5 Hz so that we could still achieve constant frequency actuation even at the highest joint oscillation amplitudes.

-“ discretizing” -> discretizing?

We thank the reviewer for the attention to detail and have fixed this (page 13, paragraph 6).

-“ The maximum speed of the simulated robot in either of these configurations with two body segments was 5% of the maximum speed with three body segments.” Would like to see more information about these experiments, with comprehensive results, in an SI.

We thank the reviewer for this suggestion and have added these results in Supplementary Table S3. We have also added some results for the simulated fish-scale robot which indicate that the deactivation of one joint is more significant when the robot has a larger mass and inertia.

Table S3: Speed of the robot-scale simulation with frozen joint configurations

Scenario	Fish scale (2.5 Hz)	Robot scale (0.5 Hz)
Base gait	0.334 BL/s	0.261 BL/s
Head amplitude = 0	-0.049 BL/s	-0.024 BL/s
Tail amplitude = 0	0.083 BL/s	0.016 BL/s

We have also added Supplementary Movie S7 that illustrates these cases.

-"The first variation is with all three segments of equal length (1:1:1 ratio of the segment lengths)." Would be clearer to specify that this is with respect to the original. I know it's implied, but upon first readthrough, I did each segment length was just equivalent.

We thank the reviewer for pointing out this ambiguous language and have rewritten this statement (page 14, paragraph 2):

(i.e., 1:1:1 ratio of the segment lengths such that each segment is 1/3 the total length of the base morphology)

-Fig. 8: Would be interesting to see more supernatural embodiments, since the simulation is "cheap" to run anyway. For example, why not 1:1:9? On another note, the x-axis of a) does not make sense to me – I don't see how the ratios map to scalar values on the x-axis.

We thank the reviewer for the suggestions. The values on the x-axis should have been categorical rather than numerical as we were trying to qualitatively assess how different the morphologies were from the base morphology. We have updated Fig. 8a to have a new x-axis which quantifies the difference in morphologies using the difference (Euclidean distance) between the new segment lengths and the base segment lengths. We continue to use color to denote the categorical relationships between the datapoints, labeling the base morphology in blue, the morphology with equal length segments in red, and the morphologies with a single long segment in yellow:

The updated Fig. 8 caption is as follows:

a) Speed of simulated robot with the same total body length but different body proportions. Blue denotes the base morphology (2:4:5 ratio of segment lengths), red denotes the morphology with equal segment lengths (1:1:1 ratio of segment lengths), and yellow denotes the morphologies with two short segments and one long segment (either 2:2:7, 2:7:2, or 7:2:2 ratios of segment lengths). b) Speed of simulated robot versus amplitude of the actuated fin segment for different lengths of the fin segment L_0 . Each version of the simulated robot was tested with five amplitudes of θ_1 ; points not included in the plot were cases in which the robot could not produce stable locomotion and rolled.

To describe the new x-axis, we have added a line in the text (page 14, paragraph 2):

The walking speed of the simulated robot with these morphologies is plotted against the difference (Euclidean distance) between the new segment lengths and the base morphology.

We also have added a description of the calculation of Euclidean distance to the Methods Section 4.3 (page 22, paragraph 4):

To calculate the Euclidean distance between the segment lengths (Fig. 8a), we use the following equation:

$$d = \sqrt{(s_{1,base} - s_{1,morph})^2 + (s_{2,base} - s_{2,morph})^2 + (s_{3,base} - s_{3,morph})^2}$$

where $s_{n, base}$ is the length of the n th segment of the base morphology and $s_{n, morph}$ is the length of the n th segment of the altered morphology.

We agree that additional simulations of unusual morphologies would be a nice addition to our analysis. In our work here, we have considered some alternate morphologies such as the 2-2-7 and the 7-2-2 ratios as examples of potential non-biological morphologies primarily because of the scalability to the physical system (e.g., the size of the motors). Further, more extreme supernatural morphologies than these are a separate research question and are out of the scope of the evaluation presented here, though we are confident our modeling method will extend to these cases.

-“... this experiment implies that the same gait will generate locomotion of different speeds when applied to different body morphologies.” This is not particularly surprising – is this would be curious to see more elaboration, especially discussion of the supernatural body types.

We thank the reviewer for this comment and have rephrased this to focus on the idea that the variation in the undulating tripod gait observed in different species (Fig. 4) might be directly connected to morphology (page 15, paragraph 1):

These example morphologies were chosen to span some of the potential design space of the simulated robot and support the idea that a range of species capable of various degrees of body bending would be expected to exhibit different parameters for the undulating tripod gait (Fig. 4).

Exploring extreme supernatural morphologies is a further research question and is out of the scope of this work.

-Fig. 9d-e: undefined error bars and sample sizes

We thank the reviewer for catching this omission and have added the following to the Fig. 9 caption:

Error bars denote the standard deviation above and below the mean point for $n = 5$ trials.

-Best to cite coefficients of friction rather than just saying ‘low’ and ‘high’

We thank the reviewer for the comment and have measured the friction coefficients of the four cases and have added the information to the Fig. 9 caption:

f) Comparison of the speed of the robot on different surfaces. “Base” denotes the robot with high friction fins on the low friction surface ($\mu_s = 0.30$), but the robot

was not able to walk with the high friction fins on the high friction surface ($\mu_s = 0.38$). "Low fr fin" denotes the robot with low friction fins on the low friction surface ($\mu_s = 0.27$), and "high fr surface" denotes the robot with low friction fins on the high friction surface ($\mu_s = 0.32$).

-From the video, it looks like the experimental robot struggles to move straight – do you change calculation of the velocity for these deviating trials?

We thank the reviewer for the comment. Because of the initial condition, the robot does move in a diagonal trajectory within the frame. However, the path of the robot is quite consistent, though we appreciate that this is difficult to see from the video provided. Speed is calculated the same way for the biological results, the simulated robot results, and the physical robot results. We take the Euclidean distance between the final position and the initial position divided by body length and time.

To clarify this, we have added the following to the Methods section 4.1 (page 21, paragraph 1):

To calculate the walking speed of *P. senegalus*, we divided the Euclidean distance traveled by the body length of the specimen and the time over which it walked.

We have added the following to the Methods section 4.3 (page 22, paragraph 2):

We used the same method to calculate the walking speed of the simulated robot as we did with the walking speed of *P. senegalus*: we divided the Euclidean distance traveled by the body length of the specimen and the time over which it walked.

We have added the following to the Methods section 4.4 (page 23, paragraph 2):

We used the same method to calculate the walking speed of the physical robot as we did with the walking speed of *P. senegalus* and the simulated robot: we divided the Euclidean distance traveled by the body length of the specimen and the time over which it walked.

We also have Supplementary Figure S6 that demonstrates the consistency of the robot's locomotion and have updated the figure caption:

Figure S6: X-Y trajectory of the head of the physical robot (n = 5 trials, 3 gait cycles each) with the fastest gait ($f = 0.5$ Hz, $A_2 = 1.5$, $A_3 = 2.1$, $\phi_2 = \pi/2$, $\phi_3 = \pi$).

-“ Unlike traditional kinematic analyses, which only observe how animals move, physics-based robotic models can test how and why certain motions lead to successful locomotion under well-defined constraints [17]. In particular...” Seems like a strange place to justify the methodology, in the discussion. Also seems redundant.

We thank the reviewer for this comment and have removed the sentence.

Reviewer #3 (Remarks on code availability):

A little more organization for the repo would help with reproducibility; why are there "robot" and "fish" directories with the same readme.tex in each?

We thank the reviewer for providing feedback on the repo. The mistake with the copied readme files has been corrected to show the correct file names. There is a “fish” directory for the experiments at the scale of *P. senegalus* (~10cm) and there is a “robot” directory for the experiments at the scale of the physical robot (~40cm).

We have also added a readme file to the top folder of the repo to provide more information about how to use the repo and what parameters to change in the MATLAB script to change the gait parameters.

The updated repo can be found here:

https://github.com/mishidasan/undulating_tripod_gait_simulink_rev1

REVIEWERS' COMMENTS

Reviewer #1 (Remarks to the Author):

I appreciate the author's work to address my comments. The changes (or justifications in a few of the minor comments) are thorough and I no longer have any major comments. There are a few minor points that the authors should address listed below, but they are easily addressed and don't change anything substantive about the paper. The manuscript as it now stands is convincing of the importance of the undulating tripod gait in the evolution of terrestrial locomotion in fishes. It does an excellent job of dissecting the behavior through multiple evidentiary avenues, and is an excellent step towards understanding the biomechanical basis that is necessary for this locomotor style as well as how variation in morphology and kinematics affect it.

Minor comments:

Abstract:

“In this work, we model of the undulating tripod gait” should be “...we model the undulating...”

We agree and have made the change as suggested.

Intro:

The first two paragraphs are a bit confusing to read. The narrative switches between terrestrial and aquatic locomotion several times, while also switching between focusing on convergent and divergent morphology/kinematics.

Here, we are trying to discuss both terrestrial and aquatic locomotion of fish at the same time because the fish are creating both walking and swimming using the same morphology. We are attempting to contrast the way current literature describe various fish species' walking gaits with swimming gaits – on a more continuous basis versus a more categorical basis.

To address this, we revised this paragraph to add some transitional words between the paragraphs and to link the walking and swimming together more clearly.

Methods:

I'm aware that other reviewers asked for additional details in the methods, which is why the authors expanded these sections. While I agree that this additional information is important to make available, I'm not sure that it should all be in the main text. I think the

previous version felt more succinct and better fit the format of the journal. As such, I would gently suggest moving the following to a supplemental text file:

- Specifics of the DeepLabCut methodology
- Specifics of the midline tracking methodology
- Explanation of the walking speed measurement for the simulation and robot
- Formula for the Euclidean distance between segment lengths used in Fig. 8a

We agree in principle, but the editorial staff has asked other methods to be moved out of Supplementary to the main Methods section so we will leave the text where it is unless directly instructed otherwise.

Figures:

Figure 8. I appreciate that the authors made the x-axis of 8a more quantitative. However, I don't think the new axis label is informatively titled. "Euclidean distance" is the method, but doesn't tell you what the meaning of the measurement is. Probably better to label it "total difference from base morphology (mm)" or something like that.

We agree and have made the change as suggested.

Reviewer #1 (Remarks on code availability):

I don't have simulink, so I was not able to run the code. However, the documentation for the code seemed appropriate.

Reviewer #2 (Remarks to the Author):

I appreciate the efforts that the authors have gone through to address the comments of myself and the other reviewers and I'm satisfied that their revisions sufficiently address said comments.

Reviewer #3 (Remarks to the Author):

The authors addressed my comments quite thoroughly. I am happy to see major improvements in the quality of the manuscript. In particular, having resolved text ambiguities, collected additional data points for experimental plots, improved methodological details for reproducibility, and explicitly acknowledged limitations of the present analysis, the paper stands as much stronger. It is my recommendation to proceed with acceptance.

I have two minor comments.

-I think Fig 2 could benefit from coordinate systems, such that it is absolutely clear about the orientation of the “top” and “front” fish views relative to one another.

We believe that the best way to describe the model is with references to features of the body rather than a coordinate system in the world or body frames. The anatomical references of the head, the tail, the dorsal (top) view, and anterior (front) view are common biological references as used in this figure and this manuscript to describe the model.

-grammatical subtlety in the abstract: “In this work, we model of the undulating tripod gait by approximating the fish’s axial undulation...” -> I wouldn’t use “the” because you have yet to specify what type of fish. Might be best to say you model “a fish’s” or “fishes’” and then agree the subsequent verb endings.

We agree and have changed “the fish’s” to “a fish’s” as suggested.

Reviewer #3 (Remarks on code availability):

The authors fixed the previous issue with the same README file in each directory; the code structure and instructions now appear just fine.